# Effective Regularization Through Loss-Function Metalearning

## Abstract

Loss-function metalearning can be used to discover novel, customized loss functions for deep neural networks, resulting in improved performance, faster training, and improved data utilization. A likely explanation is that such functions discourage overfitting, leading to effective regularization. This paper demonstrates theoretically that this is indeed the case for the TaylorGLO method: Decomposition of learning rules makes it possible to characterize the training dynamics and show that the loss functions evolved by TaylorGLO balance the pull to zero error, and a push away from it to avoid overfitting. This observation leads to an invariant that can be utilized to make the metalearning process more efficient in practice, and result in networks that are robust against adversarial attacks. Loss-function optimization can thus be seen as a well-founded new aspect of metalearning in neural networks.

## 1 Introduction

Regularization is a key concept in deep learning: it guides learning towards configurations that are likely to perform robustly on unseen data. Different regularization approaches originate from intuitive understanding of the learning process and have been shown to be effective empirically. However, the understanding of the underlying mechanisms, the different types of regularization, and their interactions, is limited.

Recently, loss function optimization has emerged as a new area of metalearning, and shown great potential in training better models. Experiments suggest that metalearned loss functions serve as regularizers in a surprising but transparent way: they prevent the network from learning too confident predictions (e.g. Baikal loss; Gonzalez & Miikkulainen, 2020a). While it may be too early to develop a comprehensive theory of regularization, given the relatively nascent state of this area, it may be possible to make progress in understanding regularization of this specific type. That is the goal of this paper.

Since metalearned loss functions are customized to a given architecture-task pair, there needs to be a shared framework under which loss functions can be analyzed and compared. The TaylorGLO (Gonzalez & Miikkulainen, 2020b) technique for loss function metalearning lends itself well to such analysis: It represents loss functions as multivariate Taylor polynomials, and leverages evolution to optimize a fixed number of parameters in this representation. In this framework, the SGD learning rule is decomposed to coefficient expressions that can be defined for a wide range of loss functions. These expressions provide an intuitive understanding of the training dynamics in specific contexts.

Using this framework, mean squared error (MSE), cross-entropy, Baikal, and TaylorGLO loss functions are analyzed at the null epoch, when network weights are similarly distributed (Appendix C), and in a zero training error regime, where the training samples' labels have been perfectly memorized. For any intermediate point in the training process, the strength of the zero training error regime as an attractor is analyzed and a constraint on this property is derived on TaylorGLO parameters by characterizing how the output distribution's entropy changes. In a concrete TaylorGLO loss function that has been metalearned, these attraction dynamics are calculated for individual samples at every epoch in a real training run, and contrasted with those for the cross-entropy loss. This comparison provides clarity on how TaylorGLO avoids becoming overly confident in its predictions. Further, the analysis shows (in Appendix D.2) how label smoothing (Szegedy et al., 2016), a traditional type

of regularization, can be implicitly encoded by TaylorGLO loss functions: Any representable loss function has label-smoothed variants that are also representable by the parameterization.

From these analyses, practical opportunities arise. First, at the null epoch, where the desired behavior can be characterized clearly, an invariant can be derived on a TaylorGLO loss function's parameters that must hold true for networks to be trainable. This constraint is then applied within the TaylorGLO algorithm to guide the search process towards good loss functions more efficiently. Second, loss-function-based regularization results in robustness that should e.g. make them more resilient to adversarial attacks. This property is demonstrated experimentally by incorporating adversarial robustness as an objective within the TaylorGLO search process. Thus, loss-function metalearning can be seen as a well-founded and practical approach to effective regularization in deep learning.

## 2 BACKGROUND

Regularization traditionally refers to methods for encouraging smoother mappings by adding a regularizing term to the objective function, i.e., to the loss function in neural networks. It can be defined more broadly, however, e.g. as "any modification we make to a learning algorithm that is intended to reduce its generalization error but not its training error" (Goodfellow et al., 2015). To that end, many regularization techniques have been developed that aim to improve the training process in neural networks. These techniques can be architectural in nature, such as Dropout (Srivastava et al., 2014) and Batch Normalization (Ioffe & Szegedy, 2015), or they can alter some aspect of the training process, such as label smoothing (Szegedy et al., 2016) or the minimization of a weight norm (Hanson & Pratt, 1989). These techniques are briefly reviewed in this section, providing context for loss-function metalearning.

### 2.1 IMPLICIT BIASES IN OPTIMIZERS

It may seem surprising that overparameterized neural networks are able to generalize at all, given that they have the capacity to memorize a training set perfectly, and in fact sometimes do (i.e., zero training error is reached). Different optimizers have different implicit biases that determine which solutions are ultimately found. These biases are helpful in providing implicit regularization to the optimization process (Neyshabur et al., 2015). Such implicit regularization is the result of a network norm—a measure of complexity—that is minimized as optimization progresses. This is why models continue to improve even after training set has been memorized (i.e., the training error global optima is reached) (Neyshabur et al., 2017).

For example, the process of stochastic gradient descent (SGD) itself has been found to provide regularization implicitly when learning on data with noisy labels (Blanc et al., 2020). In overparameterized networks, adaptive optimizers find very different solutions than basic SGD. These solutions tend to have worse generalization properties, even though they tend to have lower training errors (Wilson et al., 2017).

### 2.2 REGULARIZATION APPROACHES

While optimizers may minimize a network norm implicitly, regularization approaches supplement this process and make it explicit. For example, a common way to restrict the parameter norm explicitly is through weight decay. This approach discourages network complexity by placing a cost on weights (Hanson & Pratt, 1989).

Generalization and regularization are often characterized at the end of training, i.e. as a behavior that results from the optimization process. Various findings have influenced work in regularization. For example, flat landscapes have better generalization properties (Keskar et al., 2017; Li et al., 2018; Chaudhari et al., 2019). In overparameterized cases, the solutions at the center of these landscapes may have zero training error (i.e., perfect memorization), and under certain conditions, zero training error empirically leads to lower generalization error (Belkin et al., 2019; Nakkiran et al., 2019). However, when a training loss of zero is reached, generalization suffers (Ishida et al., 2020). This behavior can be thought of as overtraining, and techniques have been developed to reduce it at the end of the training process, such as early stopping (Morgan & Bourlard, 1990) and flooding (Ishida et al., 2020).

Both flooding and early stopping assume that overfitting happens at the end of training, which is not always true (Golatkar et al., 2019). In fact, the order in which easy-to-generalize and hard-to-generalize concepts are learned is important for the network's ultimate generalization. For instance, larger learning rates early in the training process often lead to better generalization in the final model (Li et al., 2019). Similarly, low-error solutions found by SGD in a relatively quick manner—such as through high learning rates—often have good generalization properties (Yao et al., 2007).

Other techniques tackle overfitting by making it more difficult. Dropout (Srivastava et al., 2014) makes some connections disappear. Cutout (DeVries & Taylor, 2017), Mixup (Zhang et al., 2018), and their composition, CutMix (Yun et al., 2019), augment training data with a broader variation of examples.

Notably, regularization is not a one-dimensional continuum. Different techniques regularize in different ways that often interact. For example, flooding invalidates performance gains from early stopping (Ishida et al., 2020). However, ultimately all regularization techniques alter the gradients that result from the training loss. This observation suggests loss-function optimization might be an effective way to regularize the training process.

## 2.3 LOSS-FUNCTION METALEARNING

The idea of metalearning loss-functions has a deep history with many recent developments that have shown promise in practical settings.

Prior work in reinforcement learning showed that metalearning various types of objectives is useful. For instance, in evolving policy gradients (Houthooft et al., 2018), the policy loss is not represented symbolically, but rather as a neural network that convolves over a temporal sequence of context vectors. In reward function search (Niekum et al., 2010), the task is framed as a genetic programming problem, leveraging PushGP (Spector et al., 2001). Various actor-critic reinforcement learning approaches have tackled learning a meta-critic neural network that can generate losses (Sung et al., 2017; Zhou et al., 2020). Metalearned critic network techniques have also been applied outside of reinforcement learning to train better few-shot classifiers (Antoniou & Storkey, 2019).

In unsupervised representation learning, weight update rules for semi-supervised learning have themselves been metalearned successfully (Metz et al., 2018). The update rules were constrained to fit a biological neuron model and transferred well between tasks.

Concrete loss-function metalearning for deep networks was first introduced by Gonzalez & Miikkulainen (2020a) as an automatic way to find customized loss functions that aim to optimize a performance metric for a model. The technique, a genetic programming approach, named GLO, discovered one particular loss function, Baikal, that improves classification accuracy, training speed, and data utilization. Baikal appeared to achieve these properties through a form of regularization that ensured the model would not become overly confident in its predictions. That is, instead of monotonically decreasing the loss when the output gets closer to the correct value, Baikal loss increases rapidly when the output is almost correct, thus discouraging extreme accuracy. This paper shows how training dynamics are specifically impacted in this manner when training with the Baikal loss.

Overall, GLO demonstrated that while learned loss functions' generalization effects transfer across datasets and models to some extent, they are most powerful when they are customized to individual tasks and architectures. Different loss functions can take advantage of the different characteristics of each such setting. Other techniques have advanced this new field further, for example by metalearning state-dependent loss functions for inverse dynamics models (Morse et al., 2020), and using a trained network that is itself a metalearned loss function (Bechtle et al., 2019).

One particular technique, TaylorGLO (Gonzalez & Miikkulainen, 2020b), lends itself well to analyzing what makes loss-function metalearning effective. TaylorGLO represents loss functions as parameterizations of multivariate Taylor polynomials. This parameterization allows it to scale to models with millions of trainable parameters, including a variety of deep learning architectures in image classification tasks. TaylorGLO loss functions have a tunable complexity based on the order of the polynomial; third-order functions were identified to work best in practical settings.

The third-order TaylorGLO loss function parameterization provides a fixed representation that can be used to analyze a large family of loss functions through a unified methodology. In prior work, TaylorGLO loss functions were shown to improve generalization empirically (Gonzalez & Miikkulainen, 2020b). This paper complements that work, aiming to explain why that is the case by analyzing the dynamics of training theoretically.

## 3 LEARNING RULE DECOMPOSITION

This section develops the framework for the analysis in this paper. By decomposing the learning rules under different loss functions, comparisons can be drawn at different stages of the training process. Consider the standard SGD update rule:

$$\boldsymbol{\theta} \leftarrow \boldsymbol{\theta} - \eta \nabla_{\boldsymbol{\theta}} \left( \mathcal{L}(\boldsymbol{x}_i, \boldsymbol{y}_i, \boldsymbol{\theta}) \right). \tag{1}$$

where $\eta$ is the learning rate, $\mathcal{L}(\boldsymbol{x}_i, \boldsymbol{y}_i, \boldsymbol{\theta})$ is the loss function applied to the network $h(\boldsymbol{x}_i, \boldsymbol{\theta})$, $\boldsymbol{x}_i$ is an input data sample, $\boldsymbol{y}_i$ is the $i$th sample's corresponding label, and $\boldsymbol{\theta}$ is the set of trainable parameters in the model. The update for a single weight $\theta_j$ is

$$\theta_j \leftarrow \theta_j - \eta D_{\boldsymbol{j}} \left( \mathcal{L}(\boldsymbol{x}_i, \boldsymbol{y}_i, \boldsymbol{\theta}) \right) = \theta_j - \eta \frac{\partial}{\partial s} \mathcal{L}(\boldsymbol{x}_i, \boldsymbol{y}_i, \boldsymbol{\theta} + s\boldsymbol{j}) \bigg|_{s \to 0}. \tag{2}$$

where $\boldsymbol{j}$ is a basis vector for the $j$th weight. The following text illustrates decompositions of this general learning rule in a classification context for a variety of loss functions: mean squared error (MSE), the cross-entropy loss function, the general third-order TaylorGLO loss function, and the Baikal loss function. The TaylorGLO and Baikal loss functions are referenced in Section 2.3. Each decomposition results in a learning rule of the form

$$\theta_j \leftarrow \theta_j + \eta \frac{1}{n} \sum_{k=1}^{n} \left[ \gamma_k(\boldsymbol{x}_i, \boldsymbol{y}_i, \boldsymbol{\theta}) D_{\boldsymbol{j}} \left( h_k(\boldsymbol{x}_i, \boldsymbol{\theta}) \right) \right], \tag{3}$$

where $\gamma_k(\boldsymbol{x}_i, \boldsymbol{y}_i, \boldsymbol{\theta})$ is an expression that is specific to each loss function.

Substituting the **Mean squared error (MSE)** loss into Equation 2,

$$\theta_j \leftarrow \theta_j - \eta \frac{1}{n} \sum_{k=1}^{n} \left[ 2 \left( h_k(\boldsymbol{x}_i, \boldsymbol{\theta} + s\boldsymbol{j}) - y_{ik} \right) \frac{\partial}{\partial s} h_k(\boldsymbol{x}_i, \boldsymbol{\theta} + s\boldsymbol{j}) \right] \bigg|_{s \to 0} \tag{4}$$

and breaking up the coefficient expressions into $\gamma_k(\boldsymbol{x}_i, \boldsymbol{y}_i, \boldsymbol{\theta})$ results in the weight update step

$$\gamma_k(\boldsymbol{x}_i, \boldsymbol{y}_i, \boldsymbol{\theta}) = 2y_{ik} - 2h_k(\boldsymbol{x}_i, \boldsymbol{\theta}). \tag{5}$$

Substituting the **Cross-entropy loss** into Equation 2,

$$\theta_j \leftarrow \theta_j + \eta \frac{1}{n} \sum_{k=1}^{n} \left[ y_{ik} \frac{1}{h_k(\boldsymbol{x}_i, \boldsymbol{\theta} + s\boldsymbol{j})} \frac{\partial}{\partial s} h_k(\boldsymbol{x}_i, \boldsymbol{\theta} + s\boldsymbol{j}) \right] \bigg|_{s \to 0} \tag{6}$$

and breaking up the coefficient expressions into $\gamma_k(\boldsymbol{x}_i, \boldsymbol{y}_i, \boldsymbol{\theta})$ results in the weight update step

$$\gamma_k(\boldsymbol{x}_i, \boldsymbol{y}_i, \boldsymbol{\theta}) = \frac{y_{ik}}{h_k(\boldsymbol{x}_i, \boldsymbol{\theta})}. \tag{7}$$

Substituting the **Baikal loss** into Equation 2,

$$\theta_j \leftarrow \theta_j + \eta \frac{1}{n} \sum_{k=1}^{n} \left[ \left( \frac{1}{h_k(\boldsymbol{x}_i, \boldsymbol{\theta} + s\boldsymbol{j})} + \frac{y_{ik}}{h_k(\boldsymbol{x}_i, \boldsymbol{\theta} + s\boldsymbol{j})^2} \right) \frac{\partial}{\partial s} h_k(\boldsymbol{x}_i, \boldsymbol{\theta} + s\boldsymbol{j}) \right] \bigg|_{s \to 0} \tag{8}$$

and breaking up the coefficient expressions into $\gamma_k(\boldsymbol{x}_i, \boldsymbol{y}_i, \boldsymbol{\theta})$ results in the weight update step

$$\gamma_k(\boldsymbol{x}_i, \boldsymbol{y}_i, \boldsymbol{\theta}) = \frac{1}{h_k(\boldsymbol{x}_i, \boldsymbol{\theta})} + \frac{y_{ik}}{h_k(\boldsymbol{x}_i, \boldsymbol{\theta})^2}. \tag{9}$$

Substituting the **Third-order TaylorGLO loss** with parameters $\boldsymbol{\lambda}$ into Equation 2,

$$\begin{aligned} \theta_j \leftarrow \theta_j + \eta \frac{1}{n} \sum_{k=1}^{n} \bigg[ &\lambda_2 \frac{\partial}{\partial s} h_k(\boldsymbol{x}_i, \boldsymbol{\theta} + s\boldsymbol{j}) + \lambda_3 2 \left( h_k(\boldsymbol{x}_i, \boldsymbol{\theta} + s\boldsymbol{j}) - \lambda_1 \right) \frac{\partial}{\partial s} h_k(\boldsymbol{x}_i, \boldsymbol{\theta} + s\boldsymbol{j}) \\ &+ \lambda_4 3 \left( h_k(\boldsymbol{x}_i, \boldsymbol{\theta} + s\boldsymbol{j}) - \lambda_1 \right)^2 \frac{\partial}{\partial s} h_k(\boldsymbol{x}_i, \boldsymbol{\theta} + s\boldsymbol{j}) + \lambda_5 (y_{ik} - \lambda_0) \frac{\partial}{\partial s} h_k(\boldsymbol{x}_i, \boldsymbol{\theta} + s\boldsymbol{j}) \\ &+ \left( \lambda_6 (y_{ik} - \lambda_0) 2 \left( h_k(\boldsymbol{x}_i, \boldsymbol{\theta} + s\boldsymbol{j}) - \lambda_1 \right) + \lambda_7 (y_{ik} - \lambda_0)^2 \right) \frac{\partial}{\partial s} h_k(\boldsymbol{x}_i, \boldsymbol{\theta} + s\boldsymbol{j}) \bigg] \bigg|_{s \to 0} \end{aligned} \tag{10}$$

and breaking up the coefficient expressions into $\gamma_k(\boldsymbol{x}_i, \boldsymbol{y}_i, \boldsymbol{\theta})$ results in the weight update step

$$\gamma_k(\boldsymbol{x}_i, \boldsymbol{y}_i, \boldsymbol{\theta}) = 2\lambda_3 h_k(\boldsymbol{x}_i, \boldsymbol{\theta}) - 2\lambda_1\lambda_3 + 2\lambda_6 h_k(\boldsymbol{x}_i, \boldsymbol{\theta})y_{ik} - 2\lambda_6\lambda_0 h_k(\boldsymbol{x}_i, \boldsymbol{\theta})$$

$$-2\lambda_1\lambda_6 y_{ik} + 2\lambda_1\lambda_6\lambda_0 + \lambda_2 + \lambda_5 y_{ik} - \lambda_5\lambda_0 + \lambda_7 y_{ik}^2 - 2\lambda_7\lambda_0 y_{ik} \qquad (11)$$

$$+\lambda_7\lambda_0^2 + 3\lambda_4 h_k(\boldsymbol{x}_i, \boldsymbol{\theta})^2 - 6\lambda_1\lambda_4 h_k(\boldsymbol{x}_i, \boldsymbol{\theta}) + 3\lambda_4\lambda_1^2.$$

To simplify analysis, $\gamma_k(\boldsymbol{x}_i, \boldsymbol{y}_i, \boldsymbol{\theta})$ can be decomposed into a linear combination of $[1, h_k(\boldsymbol{x}_i, \boldsymbol{\theta}), h_k(\boldsymbol{x}_i, \boldsymbol{\theta})^2, h_k(\boldsymbol{x}_i, \boldsymbol{\theta})y_{ik}, y_{ik}, y_{ik}^2]$ with respective coefficients $[c_1, c_h, c_{hh}, c_{hy}, c_y, c_{yy}]$ whose values are implicitly functions of $\boldsymbol{\lambda}$:

$$\gamma_k(\boldsymbol{x}_i, \boldsymbol{y}_i, \boldsymbol{\theta}) = c_1 + c_h h_k(\boldsymbol{x}_i, \boldsymbol{\theta}) + c_{hh} h_k(\boldsymbol{x}_i, \boldsymbol{\theta})^2 + c_{hy} h_k(\boldsymbol{x}_i, \boldsymbol{\theta})y_{ik} + c_y y_{ik} + c_{yy} y_{ik}^2. \quad (12)$$

## 4 CHARACTERIZING TRAINING DYNAMICS

Using the decomposition framework above, it is possible to characterize and compare training dynamics under different loss functions. In this section, the decompositions are first analyzed under a zero training error regime to identify optimization biases that lead to implicit regularization. Second, generalizing to the entire training process, a theoretical constraint is derived on the entropy of a network's outputs. Combined with experimental data, this constraint characterizes the data fitting and regularization processes that result from the TaylorGLO training process. This characterization shows how individual training samples are treated by different loss functions.

### 4.1 OPTIMIZATION BIASES IN THE ZERO TRAINING ERROR REGIME

Certain biases in optimization imposed by a loss function can be best observed in the case where there is nothing new to learn from the training data. Consider the case where there is zero training error, that is, $h_k(\boldsymbol{x}_i, \boldsymbol{\theta}) - y_{ik} = 0$. In this case, all $h_k(\boldsymbol{x}_i, \boldsymbol{\theta})$ can be substituted with $y_{ik}$ in $\gamma_k(\boldsymbol{x}_i, \boldsymbol{y}_i, \boldsymbol{\theta})$, as is done below for the different loss functions.

**Mean squared error (MSE):** In this case,

$$\gamma_k(\boldsymbol{x}_i, \boldsymbol{y}_i, \boldsymbol{\theta}) = 2y_{ik} - 2h_k(\boldsymbol{x}_i, \boldsymbol{\theta}) = 0. \qquad (13)$$

Thus, there are no changes to the weights of the model once error reaches zero. This observation contrasts with the findings in Blanc et al. (2020), who discovered an implicit regularization effect when training with MSE loss *and* label noise. Notably, this null behavior is representable in a non-degenerate TaylorGLO parameterization, since MSE is itself representable by TaylorGLO with $\boldsymbol{\lambda} = [0, 0, 0, -1, 0, 2, 0, 0]$. Thus, this behavior can be leveraged in evolved loss functions.

**Cross-entropy loss:** Since $h_k(\boldsymbol{x}_i, \boldsymbol{\theta}) = 0$ for non-target logits in a zero training error regime, $\gamma_k(\boldsymbol{x}_i, \boldsymbol{y}_i, \boldsymbol{\theta}) = \frac{0}{0}$, i.e. an indeterminate form. Thus, an arbitrarily-close-to-zero training error regime is analyzed instead, such that $h_k(\boldsymbol{x}_i, \boldsymbol{\theta}) = \epsilon$ for non-target logits for an arbitrarily small $\epsilon$. Since all scaled logits sum to 1, $h_k(\boldsymbol{x}_i, \boldsymbol{\theta}) = 1 - (n-1)\epsilon$ for the target logit. Let us analyze the learning rule as $\epsilon$ tends towards 0:

$$\theta_j \leftarrow \theta_j + \lim_{\epsilon \to 0} \eta \frac{1}{n} \sum_{k=1}^{n} \begin{cases} \dfrac{y_{ik}}{\epsilon} D_{\boldsymbol{j}}\left(h_k(\boldsymbol{x}_i, \boldsymbol{\theta})\right) & y_{ik} = 0 \\ \dfrac{y_{ik}}{1-(n-1)\epsilon} D_{\boldsymbol{j}}\left(h_k(\boldsymbol{x}_i, \boldsymbol{\theta})\right) & y_{ik} = 1 \end{cases} \qquad (14)$$

$$= \theta_j + \eta \frac{1}{n} \sum_{k=1}^{n} \begin{cases} 0 & y_{ik} = 0 \\ D_{\boldsymbol{j}}\left(h_k(\boldsymbol{x}_i, \boldsymbol{\theta})\right) & y_{ik} = 1. \end{cases} \qquad (15)$$

Intuitively, this learning rule aims to increase the value of the target scaled logits. Since logits are scaled by a softmax function, increasing the value of one logit decreases the values of other logits. Thus, the fixed point of this bias will be to force non-target scaled logits to zero, and target scaled logits to one. In other words, this behavior aims to minimize the divergence between the predicted distribution and the training data's distribution.

TaylorGLO can represent this behavior, and can thus be leveraged in evolved loss functions, through any case where $a = 0$ and $b + c > 0$. Any $\boldsymbol{\lambda}$ where $\lambda_2 = 2\lambda_1\lambda_3 + \lambda_5\lambda_0 - 2\lambda_1\lambda_6\lambda_0 - \lambda_7\lambda_0^2 - 3\lambda_4\lambda_1^2$ represents such a satisfying family of cases. Additionally, TaylorGLO allows for the strength of this bias to be tuned independently from $\eta$ by adjusting the magnitude of $b + c$.

**Baikal loss:**    Notably, the Baikal loss function results in infinite gradients at zero training error, rendering it unstable, even if using it to fine-tune from a previously trained network that already reached zero training error. However, the zero-error regime is irrelevant with Baikal because it cannot be reached in practice:

*Theorem 1: Zero training error regions of the weight space are not attractors for the Baikal loss function.*

The reason is that if a network reaches reaches a training error that is arbitrarily close to zero, there is a repulsive effect that biases the model's weights away from zero training error. Proof of this theorem is in Appendix D.1.

**Third-order TaylorGLO loss:**    According to Equation 12, in the zero-error regime $\gamma_k(\boldsymbol{x}_i, \boldsymbol{y}_i, \boldsymbol{\theta})$ can be written as a linear combination of $[1, y_{ik}, y_{ik}^2]$ and $[a, b, c]$ (as defined in Appendix B), i.e. $\gamma_k(\boldsymbol{x}_i, \boldsymbol{y}_i, \boldsymbol{\theta}) = a + b y_{ik} + c y_{ik}^2$.

Notably, in the basic classification case, $\forall w \in \mathbb{N}_1 : y_{ik} = y_{ik}^w$, since $y_{ik} \in \{0, 1\}$. This observation provides an intuition for why higher-order TaylorGLO loss functions are not able to provide fundamentally different behavior (beyond a more overparameterized search space), and thus no improvements in performance over third-order loss functions. The learning rule thus becomes

$$\theta_j \leftarrow \theta_j + \eta \frac{1}{n} \sum_{k=1}^{n} \left\{ \begin{array}{ll} a D_{\boldsymbol{j}}\left(h_k(\boldsymbol{x}_i, \boldsymbol{\theta})\right) & y_{ik} = 0 \\ (a + b + c) D_{\boldsymbol{j}}\left(h_k(\boldsymbol{x}_i, \boldsymbol{\theta})\right) & y_{ik} = 1, \end{array} \right. \tag{16}$$

where $a, b, c$ are functions of $\boldsymbol{\lambda}$ defined in Appendix B.

As a concrete example, consider the loss function TaylorGLO discovered for the AllCNN-C model on CIFAR-10 (Gonzalez & Miikkulainen, 2020b). It had $a = -373.917, b = -129.928, c = -11.3145$. Notably, all three coefficients are negative, i.e. all changes to $\theta_j$ are a negatively scaled values of $D_{\boldsymbol{j}}\left(h_k(\boldsymbol{x}_i, \boldsymbol{\theta})\right)$, as can be seen from Equation 16. Thus, there are two competing processes in this learning rule: one that aims to minimize all non-target scaled logits (increasing the scaled logit distribution's entropy), and one that aims to minimize the target scaled logit (decreasing the scaled logit distribution's entropy). The processes conflict with each other since logits are scaled through a softmax function. These processes can shift weights in a particular way while maintaining zero training error, which results in implicit regularization. If, however, such shifts in this zero training error regime do lead to misclassifications on the training data, $h_k(\boldsymbol{x}_i, \boldsymbol{\theta})$ would no longer equal $y_{ik}$, and a non-zero error regime's learning rule would come into effect. It would strive to get back to zero training error with a different $\boldsymbol{\theta}$.

Similarly to Baikal loss, a training error of exactly zero is not an attractor for some third-order TaylorGLO loss functions (this property can be seen through an analysis similar to that in Section D.1). The zero-error case would occur in practice only if this loss function were to be used to fine tune a network that truly has a zero training error. It is, however, a useful step in characterizing the behavior of TaylorGLO, as will be seen later.

## 4.2 DATA FITTING AND REGULARIZATION PROCESSES

Under what gradient conditions does a network's softmax function transition from increasing the entropy in the output distribution to decreasing it? Characterizing this behavior can yield insights on how specific training samples affect a network's trainable parameters. Let us analyze the case where all non-target logits have the same value, $\frac{\epsilon}{n-1}$, and the target logit has the value $1 - \epsilon$. That is, all non-target classes have equal probabilities.

*Theorem 2. The strength of entropy reduction is proportional to*

$$\frac{\epsilon(\epsilon - 1) \left( e^{\epsilon(\epsilon-1)(\gamma_{\neg T} - \gamma_T)} - e^{\frac{\epsilon(\epsilon-1)\gamma_T(n-1) + \epsilon\gamma_{\neg T}(\epsilon(n-3)+n-1)}{(n-1)^2}} \right)}{(\epsilon - 1) e^{\epsilon(\epsilon-1)(\gamma_{\neg T} - \gamma_T)} - \epsilon\, e^{\frac{\epsilon(\epsilon-1)\gamma_T(n-1) + \epsilon\gamma_{\neg T}(\epsilon(n-3)+n-1)}{(n-1)^2}}} \tag{17}$$

where $\gamma_{\neg T}$ is the value of $\gamma_j$ for non-target logits, and $\gamma_T$ for the target logit. Thus, values less than zero imply that entropy is increased, values greater than zero imply that it is decreased, and values equal to zero imply that there is no change. The proof is in Appendix D.3.

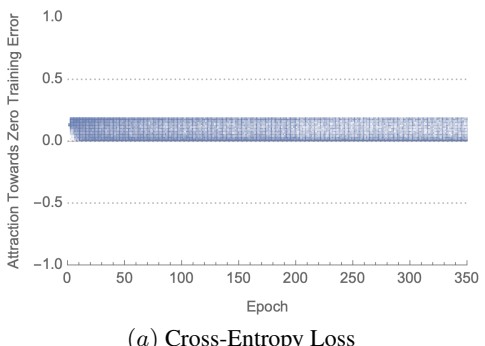

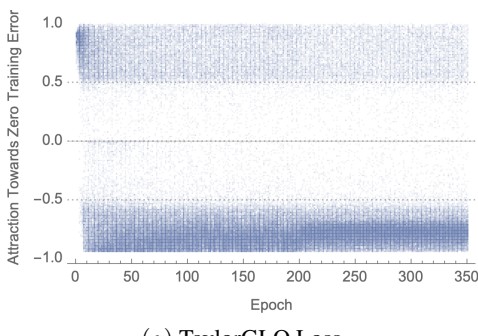

$(a)$ Cross-Entropy Loss
$(a)$ TaylorGLO Loss

Figure 1: Attraction towards zero training error with cross-entropy and TaylorGLO loss functions on CIFAR-10 AllCNN-C models. Each point represents an individual training sample (500 are randomly sampled per epoch); its $x$-location indicates the training epoch, and $y$-location the strength with which the loss functions pulls the output towards the correct label, or pushes it away from it. With the cross-entropy loss, these values are always positive, indicating a constant pull towards the correct label for every single training sample. Interestingly, the TaylorGLO values span both the positives and the negatives; at the beginning of training there is a strong pull towards the correct label (seen as the dark area on top left), which then changes to more prominent push away from it in later epochs. This plot shows how TaylorGLO regularizes by preventing overconfidence and biasing solutions towards different parts of the weight space.

The strength of the entropy reduction in Theorem 2 can also be thought of as a measure of the strength of the attraction towards zero training error regions of the parameter space (i.e., shrinking non-target logits and growing target logits imply reduced training error). This strength can be calculated for individual training samples during any part of the training process, leading to the insight that the process results from competing "push" and "pull" forces. This theoretical insight, combined with empirical data from actual training sessions, explains how different loss functions balance data fitting and regularization.

Figure 1 provides one such example on AllCNN-C (Springenberg et al., 2015) models trained on CIFAR-10 (Krizhevsky & Hinton, 2009) with cross-entropy and custom TaylorGLO loss functions. Scaled target and non-target logit values were logged for every sample at every epoch and used to calculate respective $\gamma_T$ and $\gamma_{\neg T}$ values. These values were then substituted into Equation 17 to get the strength of bias towards zero training error. Specific instances of Equation 17 for four loss functions are plotted in Appendix E.

The cross-entropy loss exhibits a tendency towards zero training error for every single sample, as expected. The TaylorGLO loss, however, has a much different behavior: initially, there is a much stronger pull towards zero training error for all samples—which leads to better generalization (Yao et al., 2007; Li et al., 2019)—after which a stratification occurs, where the majority of samples are repelled, and thus biased towards a different region of the weight space that happens to have better performance characteristics empirically. A similar behavior is identified for the Baikal loss in Appendix E.

## 5 INVARIANT ON TAYLORGLO PARAMETERS

There are many different instances of $\boldsymbol{\lambda}$ for which models are untrainable. One such case, albeit a degenerate one, is $\boldsymbol{\lambda} = \mathbf{0}$ (i.e., a function with zero gradients everywhere). Given the training dynamics at the null epoch (characterized in Appendix C), more general constraints on $\boldsymbol{\lambda}$ can be derived (in Appendix D.4), resulting in the following theorem:

*Theorem 3. A third-order TaylorGLO loss function is not trainable if the following constraints on $\boldsymbol{\lambda}$ are satisfied:*

$$c_1 + c_y + c_{yy} + \frac{c_h + c_{hy}}{n} + \frac{c_{hh}}{n^2} \quad < \quad (n-1)\left(c_1 + \frac{c_h}{n} + \frac{c_{hh}}{n^2}\right) \tag{18}$$

$$c_y + c_{yy} + \frac{c_{hy}}{n} \quad < \quad (n-2)\left(c_1 + \frac{c_h}{n} + \frac{c_{hh}}{n^2}\right). \tag{19}$$

The inverse of these constraints may be used as an invariant during loss function evolution. That is, they can be used to identify entire families of loss function parameters that are not usable, rule

Table 1: Test-set accuracy of loss functions discovered by TaylorGLO with and without an invariant constraint on $\lambda$. Models were trained on the loss function that had the highest validation accuracy during the TaylorGLO evolution. All averages are from ten separately trained models and $p$-values are from one-tailed Welch's $t$-Tests. Standard deviations are shown in parentheses. The invariant allows focusing metalearning to viable areas of the search space, resulting in better loss functions.

| Task and Model | Avg. TaylorGLO Acc. | + Invariant | $p$-value |
|---|---|---|---|
| CIFAR-10 on AlexNet [1] | 0.7901 (0.0026) | **0.7933 (0.0026)** | 0.0092 |
| CIFAR-10 on PreResNet-20 [2] | 0.9169 (0.0014) | 0.9164 (0.0019) | 0.2827 |
| CIFAR-10 on AllCNN-C [3] | 0.9271 (0.0013) | **0.9290 (0.0014)** | 0.0004 |
| CIFAR-10 on AllCNN-C [3] + Cutout [4] | 0.9329 (0.0022) | **0.9350 (0.0014)** | 0.0124 |

[1] Krizhevsky et al. (2012)  [2] He et al. (2016)  [3] Springenberg et al. (2015)  [4] DeVries & Taylor (2017)

them out during search, and thereby make the search more effective. More specifically, before each candidate $\lambda$ is evaluated, it is checked for conformance to the invariant. If the invariant is violated, the algorithm can skip that candidate's validation training and simply assign a fitness of zero. However, due to the added complexity that the invariant imposes on the fitness landscape, a larger population size is needed for evolution within TaylorGLO to be more stable. Practically, a doubling of the population size from 20 to 40 works well.

Table 1 presents results from TaylorGLO runs with and without the invariant on the CIFAR-10 image classification benchmark dataset (Krizhevsky & Hinton, 2009) with various architectures. Networks with Cutout (DeVries & Taylor, 2017) were also evaluated to show that TaylorGLO provides a different approach to regularization. Standard training hyperparameters from the references were used for each architecture. Notably, the invariant allows TaylorGLO to discover loss functions that have statistically significantly better performance in many cases and never a detrimental effect. These result demonstrate that the theoretical invariant is useful in practice, and should become a standard in TaylorGLO applications.

## 6 ADVERSARIAL ROBUSTNESS

TaylorGLO loss functions discourage overconfidence, i.e. their activations are less extreme and vary more smoothly with input. Such encodings are likely to be more robust against noise, damage, and other imperfections in the data and in the network execution. In the extreme case, they may also be more robust against adversarial attacks. This hypothesis will be tested experimentally in this section.

Adversarial attacks elicit incorrect predictions from a trained model by changing input samples in small ways that can even be imperceptible. They are generally classified as "white-box" or "black-box" attacks, depending on whether the attacker has access to the underlying model or not, respectively. Naturally white-box attacks are more powerful at overwhelming a model. One such white-box attack is the Fixed Gradient Sign Method (FGSM; Goodfellow et al., 2015): following evaluation of a dataset, input gradients are taken from the network following a backward pass. Each individual gradient has its sign calculated and scaled by an $\epsilon$ scaling factor that determines the attack strength. These values are added to future network inputs with an $\epsilon$ scaling factor, causing misclassifications.

Figure 2 shows how robust networks with different loss functions are to FGSM attacks of various strengths. In this experiment, AllCNN-C and Wide ResNet 28-5 (Zagoruyko & Komodakis, 2016) networks were trained on CIFAR-10 with TaylorGLO and cross-entropy loss; indeed TaylorGLO outperforms the cross-entropy loss models significantly at all attack strengths. Note that in this case, loss functions were evolved simply to perform well, and adversarial robustness emerged as a side benefit. However, it is also possible to take adversarial attacks into account as an explicit objective in loss function evolution. Since TaylorGLO can uses non-differentiable metrics as objectives in its search process, the traditional validation accuracy objective can be replaced with validation accuracy at a particular FGSM attack strength. Remarkably, loss functions found with this objective outperform both the previous TaylorGLO loss functions and the cross-entropy loss. These results demonstrate that the TaylorGLO regularization leads to robust encoding (detailed in Appendix F), and such robustness can be further improved by making it an explicit goal in loss-function optimization.

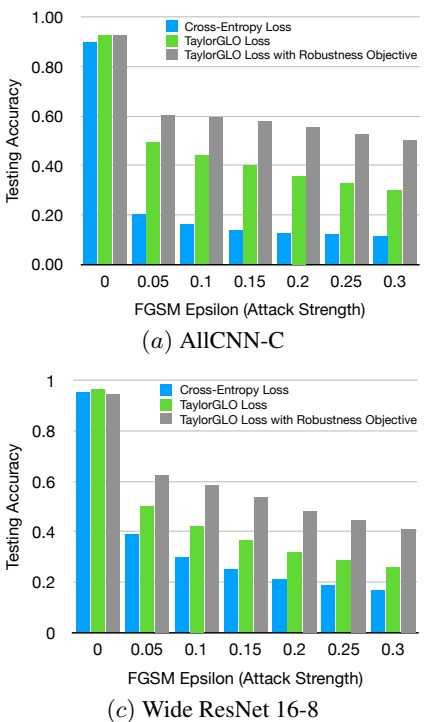

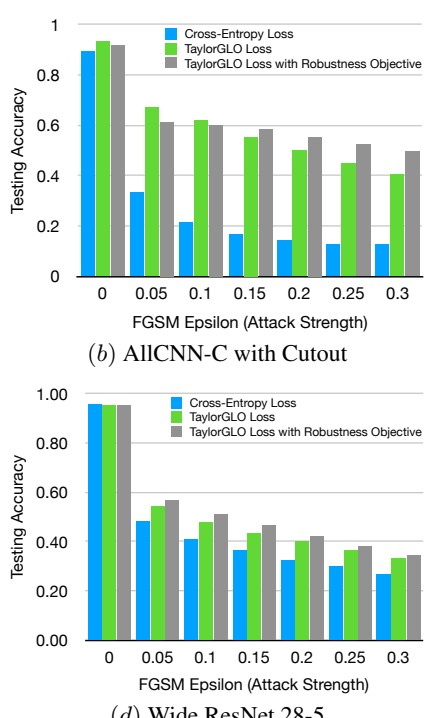

Figure 2: Robustness of TaylorGLO loss functions against FGSM adversarial attacks on CIFAR-10. For each architecture, the blue bars represent accuracy achieved through training with the cross-entropy loss, green bars with a TaylorGLO loss, and gray bars with a TaylorGLO loss specifically evolved in the adversarial attack environment. The leftmost points on each plot represent evaluations without adversarial attacks. TaylorGLO regularization makes the networks more robust against adversarial attacks, and this property can be further enhanced by making it an explicit goal in evolution.

Future work can naturally extend these analyses to black-box adversarial attacks and adversarial training. Moreover, just as custom loss functions can improve intrinsic robustness against adversarial attacks, custom loss function may make adversarial training more effective.

## 7 CONCLUSION

Regularization has long been a crucial aspect of training deep neural networks, and exists in many different flavors. This paper contributed an understanding of one recent and compelling family of regularization techniques: loss-function metalearning. A theoretical framework for representing different loss functions was first developed in order to analyze their training dynamics in various contexts. The results demonstrate that TaylorGLO loss functions implement a guard against overfitting, resulting in automatic regularization. Two practical opportunities emerged from this analysis: filtering based on an invariant was shown to improve the search process, and the robustness against overfitting to make the networks more robust against adversarial attacks. The results thus extend the scope of metalearning, focusing it not just on finding optimal model configurations, but also on improving regularization, learning efficiency, and robustness directly.

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

## A    NOTATION OVERVIEW

| Symbol | Description |
|--------|-------------|
| $h(\boldsymbol{x}_i, \boldsymbol{\theta})$ | The model, with a softmax |
| $h_k(\boldsymbol{x}_i, \boldsymbol{\theta})$ | The model's $k$th scaled logit |
| $D_{\boldsymbol{j}}(f)$ | The directional derivative of $f$ along $\boldsymbol{j}$ |
| $\mathbb{P}_{\text{data}}$ | Probability distribution of original data |
| $\boldsymbol{x}_i$ | An input data sample, where $\boldsymbol{x}_i \sim \mathbb{P}_{\text{data}}$ |
| $\boldsymbol{y}_i$ | A label that corresponds to the $\boldsymbol{x}_i$ sample |
| $\eta$ | Learning rate |
| $n$ | Number of classes |
| $\boldsymbol{\theta}$ | A model's trainable parameters |
| $\boldsymbol{\lambda}$ | The loss function's parameters |
| $\mathcal{L}(\boldsymbol{x}_i, \boldsymbol{y}_i, \boldsymbol{\theta})$ | The loss function |

## B    THIRD-ORDER TAYLORGLO LOSS FUNCTION IN THE ZERO TRAINING ERROR REGIME

$$\gamma_k(\boldsymbol{x}_i, \boldsymbol{y}_i, \boldsymbol{\theta}) = a + by_{ik} + cy_{ik}^2, \tag{20}$$

where

$$a = \lambda_2 - 2\lambda_1\lambda_3 - \lambda_5\lambda_0 + 2\lambda_1\lambda_6\lambda_0 + \lambda_7\lambda_0^2 + 3\lambda_4\lambda_1^2 \tag{21}$$
$$b = 2\lambda_3 - 2\lambda_6\lambda_0 - 2\lambda_1\lambda_6 + \lambda_5 - 2\lambda_7\lambda_0 - 6\lambda_4\lambda_1 \tag{22}$$
$$c = 2\lambda_6 + \lambda_7 + 3\lambda_4. \tag{23}$$

## C    BEHAVIOR AT THE NULL EPOCH

Consider the first epoch of training. Assume all weights are randomly initialized:

$$\forall k \in [1, n], \text{where } n \geq 2 : \mathbb{E}_i [h_k(\boldsymbol{x}_i, \boldsymbol{\theta})] = \frac{1}{n}. \tag{24}$$

That is, logits are distributed with high entropy. Behavior at the null epoch can then be defined piecewise for target vs. non-target logits for each loss function.

In the case of **Mean squared error (MSE),**

$$\gamma_k(\boldsymbol{x}_i, \boldsymbol{y}_i, \boldsymbol{\theta}) = \begin{cases} 2n^{-1} & y_{ik} = 0 \\ 2n^{-1} - 2 & y_{ik} = 1. \end{cases} \tag{25}$$

Since $n \geq 2$, the $y_{ik} = 1$ case will always be negative, while the $y_{ik} = 0$ case will always be positive. Thus, target scaled logits will be maximized and non-target scaled logits minimized.

In the case of **Cross-entropy loss,**

$$\gamma_k(\boldsymbol{x}_i, \boldsymbol{y}_i, \boldsymbol{\theta}) = \begin{cases} 0 & y_{ik} = 0 \\ n & y_{ik} = 1. \end{cases} \tag{26}$$

Target scaled logits are maximized and, consequently, non-target scaled logits minimized as a result of the softmax function.

Similarly in the case of **Baikal loss,**

$$\gamma_k(\boldsymbol{x}_i, \boldsymbol{y}_i, \boldsymbol{\theta}) = \left\{ \begin{array}{ll} n & y_{ik} = 0 \\ n + n^2 & y_{ik} = 1. \end{array} \right. \tag{27}$$

Target scaled logits are minimized and, consequently, non-target scaled logits minimized as a result of the softmax function (since the $y_{ik} = 1$ case dominates).

In the case of **Third-order TaylorGLO loss,** since behavior is highly dependent on $\boldsymbol{\lambda}$, consider the concrete loss function used above:

$$\gamma_k(\boldsymbol{x}_i, \boldsymbol{y}_i, \boldsymbol{\theta}) = \left\{ \begin{array}{ll} -373.917 - 130.264\, h_k(\boldsymbol{x}_i, \boldsymbol{\theta}) - 11.2188\, h_k(\boldsymbol{x}_i, \boldsymbol{\theta})^2 & y_{ik} = 0 \\ -372.470735 - 131.47\, h_k(\boldsymbol{x}_i, \boldsymbol{\theta}) - 11.2188\, h_k(\boldsymbol{x}_i, \boldsymbol{\theta})^2 & y_{ik} = 1. \end{array} \right. \tag{28}$$

Note that Equation 16 is a special case of this behavior where $h_k(\boldsymbol{x}_i, \boldsymbol{\theta}) = y_{ik}$. Let us substitute $h_k(\boldsymbol{x}_i, \boldsymbol{\theta}) = \frac{1}{n}$ (i.e., the expected value of a logit at the null epoch):

$$\gamma_k(\boldsymbol{x}_i, \boldsymbol{y}_i, \boldsymbol{\theta}) = \left\{ \begin{array}{ll} -373.917 - 130.264\, n^{-1} - 11.2188\, n^{-2} & y_{ik} = 0 \\ -372.470735 - 131.47\, n^{-1} - 11.2188\, n^{-2} & y_{ik} = 1. \end{array} \right. \tag{29}$$

Since this loss function was found on CIFAR-10, a 10-class image classification task, $n = 10$:

$$\gamma_k(\boldsymbol{x}_i, \boldsymbol{y}_i, \boldsymbol{\theta}) = \left\{ \begin{array}{ll} -386.9546188 & y_{ik} = 0 \\ -385.729923 & y_{ik} = 1. \end{array} \right. \tag{30}$$

Since both cases of $\gamma_k(\boldsymbol{x}_i, \boldsymbol{y}_i, \boldsymbol{\theta})$ are negative, this behavior implies that all scaled logits will be minimized. However, since the scaled logits are the output of a softmax function, and the $y_{ik} = 0$ case is more strongly negative, the non-target scaled logits will be minimized more than the target scaled logits, resulting in a maximization of the target scaled logits.

## D   PROOFS AND DERIVATIONS

Proofs and derivations for theorems in the paper are presented below:

### D.1   ZERO TRAINING ERROR IS NOT AN ATTRACTOR OF BAIKAL

Given that Baikal does tend to minimize training error to a large degree—otherwise it would be useless as a loss function since we are effectively assuming that the training data is in-distribution—we can observe what happens as we approach a point in parameter space that is arbitrarily-close to zero training error. Assume, without loss of generality, that all non-target scaled logits have the same value.

$$\theta_j \leftarrow \theta_j + \eta \frac{1}{n} \sum_{k=1}^{n} \left\{ \begin{array}{ll} \lim\limits_{h_k(\boldsymbol{x}_i, \boldsymbol{\theta}) \to \frac{\epsilon}{n-1}} \gamma_k(\boldsymbol{x}_i, \boldsymbol{y}_i, \boldsymbol{\theta}) D_{\boldsymbol{j}}\left(h_k(\boldsymbol{x}_i, \boldsymbol{\theta})\right) & y_{ik} = 0 \\ \lim\limits_{h_k(\boldsymbol{x}_i, \boldsymbol{\theta}) \to 1-\epsilon} \gamma_k(\boldsymbol{x}_i, \boldsymbol{y}_i, \boldsymbol{\theta}) D_{\boldsymbol{j}}\left(h_k(\boldsymbol{x}_i, \boldsymbol{\theta})\right) & y_{ik} = 1 \end{array} \right. \tag{31}$$

$$= \theta_j + \eta \frac{1}{n} \sum_{k=1}^{n} \left\{ \begin{array}{ll} \lim\limits_{h_k(\boldsymbol{x}_i, \boldsymbol{\theta}) \to \frac{\epsilon}{n-1}} \left( \dfrac{1}{h_k(\boldsymbol{x}_i, \boldsymbol{\theta})} + \dfrac{0}{h_k(\boldsymbol{x}_i, \boldsymbol{\theta})^2} \right) D_{\boldsymbol{j}}\left(h_k(\boldsymbol{x}_i, \boldsymbol{\theta})\right) & y_{ik} = 0 \\ \lim\limits_{h_k(\boldsymbol{x}_i, \boldsymbol{\theta}) \to 1-\epsilon} \left( \dfrac{1}{h_k(\boldsymbol{x}_i, \boldsymbol{\theta})} + \dfrac{1}{h_k(\boldsymbol{x}_i, \boldsymbol{\theta})^2} \right) D_{\boldsymbol{j}}\left(h_k(\boldsymbol{x}_i, \boldsymbol{\theta})\right) & y_{ik} = 1 \end{array} \right. \tag{32}$$

$$= \theta_j + \eta \frac{1}{n} \sum_{k=1}^{n} \left\{ \begin{array}{ll} \dfrac{n-1}{\epsilon} D_{\boldsymbol{j}}\left(h_k(\boldsymbol{x}_i, \boldsymbol{\theta})\right) & y_{ik} = 0 \\ \left( \dfrac{1}{1-\epsilon} + \dfrac{1}{(1-\epsilon)^2} \right) D_{\boldsymbol{j}}\left(h_k(\boldsymbol{x}_i, \boldsymbol{\theta})\right) & y_{ik} = 1 \end{array} \right. \tag{33}$$

$$= \theta_j + \eta \frac{1}{n} \sum_{k=1}^{n} \left\{ \begin{array}{ll} \dfrac{n-1}{\epsilon} D_{\boldsymbol{j}}\left(h_k(\boldsymbol{x}_i, \boldsymbol{\theta})\right) & y_{ik} = 0 \\ \dfrac{2-\epsilon}{\epsilon^2 - 2\epsilon + 1} D_{\boldsymbol{j}}\left(h_k(\boldsymbol{x}_i, \boldsymbol{\theta})\right) & y_{ik} = 1 \end{array} \right. \tag{34}$$

The behavior in the $y_{ik} = 0$ case will dominate for small values of $\epsilon$. Both cases have a positive range for small values of $\epsilon$, ultimately resulting in non-target scaled logits becoming maximized, and subsequently the non-target logit becoming minimized. This is equivalent, in expectation, to saying that $\epsilon$ will become larger after applying the learning rule. A larger $\epsilon$ clearly implies a move away from a zero training error area of the parameter space. Thus, zero training error is not an attractor for the Baikal loss function.

## D.2 Label smoothing in TaylorGLO

Consider a basic setup with standard label smoothing, controlled by a hyperparameter $\alpha \in (0, 1)$, such that the target value in any $\boldsymbol{y}_i$ is $1 - \alpha\frac{n-1}{n}$, rather than 1, and non-target values are $\frac{\alpha}{n}$, rather than 0. The learning rule changes in the general case as follows:

$$\gamma_k(\boldsymbol{x}_i, \boldsymbol{y}_i, \boldsymbol{\theta}) = \begin{cases} \begin{aligned} & c_1 + c_h h_k(\boldsymbol{x}_i, \boldsymbol{\theta}) + c_{hh} h_k(\boldsymbol{x}_i, \boldsymbol{\theta})^2 \\ & + c_{hy} h_k(\boldsymbol{x}_i, \boldsymbol{\theta})\frac{\alpha}{n} + c_y \frac{\alpha}{n} + c_{yy}\frac{\alpha^2}{n^2} \end{aligned} & y_{ik} = 0 \\[2em] \begin{aligned} & c_1 + c_h h_k(\boldsymbol{x}_i, \boldsymbol{\theta}) + c_{hh} h_k(\boldsymbol{x}_i, \boldsymbol{\theta})^2 + c_{hy} h_k(\boldsymbol{x}_i, \boldsymbol{\theta})\left(1 - \alpha\frac{n-1}{n}\right) \\ & + c_y\left(1 - \alpha\frac{n-1}{n}\right) + c_{yy}\left(1 - \alpha\frac{n-1}{n}\right)^2 \end{aligned} & y_{ik} = 1 \end{cases} \tag{35}$$

Let $\hat{c}_1, \hat{c}_h, \hat{c}_{hh}, \hat{c}_{hy}, \hat{c}_y, \hat{c}_{yy}$ represent settings for $c_1, c_h, c_{hh}, c_{hy}, c_y, c_{yy}$ in the non-label-smoothed case that implicitly apply label smoothing within the TaylorGLO parameterization. Given the two cases in the label-smoothed and non-label-smoothed definitions of $\gamma_k(\boldsymbol{x}_i, \boldsymbol{y}_i, \boldsymbol{\theta})$, there are two equations that must be satisfiable by settings of $\hat{c}$ constants for any $c$ constants, with shared terms highlighted in blue and red:

$$c_1 + c_h h_k(\boldsymbol{x}_i, \boldsymbol{\theta}) + c_{hh} h_k(\boldsymbol{x}_i, \boldsymbol{\theta})^2 + c_{hy} h_k(\boldsymbol{x}_i, \boldsymbol{\theta})\frac{\alpha}{n} + c_y\frac{\alpha}{n} + c_{yy}\frac{\alpha^2}{n^2} \tag{36}$$
$$= \hat{c}_1 + \hat{c}_h h_k(\boldsymbol{x}_i, \boldsymbol{\theta}) + \hat{c}_{hh} h_k(\boldsymbol{x}_i, \boldsymbol{\theta})^2$$

$$c_1 + c_h h_k(\boldsymbol{x}_i, \boldsymbol{\theta}) + c_{hh} h_k(\boldsymbol{x}_i, \boldsymbol{\theta})^2 + c_{hy} h_k(\boldsymbol{x}_i, \boldsymbol{\theta})\left(1 - \alpha\frac{n-1}{n}\right)$$
$$+ c_y\left(1 - \alpha\frac{n-1}{n}\right) + c_{yy}\left(1 - \alpha\frac{n-1}{n}\right)^2 \tag{37}$$
$$= \hat{c}_1 + \hat{c}_h h_k(\boldsymbol{x}_i, \boldsymbol{\theta}) + \hat{c}_{hh} h_k(\boldsymbol{x}_i, \boldsymbol{\theta})^2 + \hat{c}_{hy} h_k(\boldsymbol{x}_i, \boldsymbol{\theta}) + \hat{c}_y + \hat{c}_{yy}$$

Let us then factor the left-hand side of Equation 36 in terms of different powers of $h_k(\boldsymbol{x}_i, \boldsymbol{\theta})$:

$$\underbrace{\left(c_1 + c_y\frac{\alpha}{n} + c_{yy}\frac{\alpha^2}{n^2}\right)}_{\hat{c}_1} + \underbrace{\left(c_h + c_{hy}\frac{\alpha}{n}\right)}_{\hat{c}_h} h_k(\boldsymbol{x}_i, \boldsymbol{\theta}) + \underbrace{c_{hh}}_{\hat{c}_{hh}} h_k(\boldsymbol{x}_i, \boldsymbol{\theta})^2 \tag{38}$$

Resulting in definitions for $\hat{c}_1, \hat{c}_h, \hat{c}_{hh}$. Let us then add the following form of zero to the left-hand side of Equation 37:

$$\left(c_{hy} h_k(\boldsymbol{x}_i, \boldsymbol{\theta})\frac{\alpha}{n} + c_y\frac{\alpha}{n} + c_{yy}\frac{\alpha^2}{n^2}\right) - \left(c_{hy} h_k(\boldsymbol{x}_i, \boldsymbol{\theta})\frac{\alpha}{n} + c_y\frac{\alpha}{n} + c_{yy}\frac{\alpha^2}{n^2}\right) \tag{39}$$

This allows us to substitute the definitions for $\hat{c}_1, \hat{c}_h, \hat{c}_{hh}$ from Equation 38 into Equation 37:

$$\hat{c}_1 + \hat{c}_h h_k(\boldsymbol{x}_i, \boldsymbol{\theta}) + \hat{c}_{hh} h_k(\boldsymbol{x}_i, \boldsymbol{\theta})^2 - \left(c_{hy} h_k(\boldsymbol{x}_i, \boldsymbol{\theta})\frac{\alpha}{n} + c_y\frac{\alpha}{n} + c_{yy}\frac{\alpha^2}{n^2}\right)$$
$$+ c_{hy} h_k(\boldsymbol{x}_i, \boldsymbol{\theta})\left(1 - \alpha\frac{n-1}{n}\right) + c_y\left(1 - \alpha\frac{n-1}{n}\right) + c_{yy}\left(1 - \alpha\frac{n-1}{n}\right)^2 \tag{40}$$
$$= \hat{c}_1 + \hat{c}_h h_k(\boldsymbol{x}_i, \boldsymbol{\theta}) + \hat{c}_{hh} h_k(\boldsymbol{x}_i, \boldsymbol{\theta})^2 + \hat{c}_{hy} h_k(\boldsymbol{x}_i, \boldsymbol{\theta}) + \hat{c}_y + \hat{c}_{yy}$$

Simplifying into:

$$c_{hy} h_k(\boldsymbol{x}_i, \boldsymbol{\theta}) \left(1 - \alpha \frac{n-1}{n}\right) + c_y \left(1 - \alpha \frac{n-1}{n}\right) + c_{yy} \left(1 - \alpha \frac{n-1}{n}\right)^2$$
$$- \left(c_{hy} h_k(\boldsymbol{x}_i, \boldsymbol{\theta}) \frac{\alpha}{n} + c_y \frac{\alpha}{n} + c_{yy} \frac{\alpha^2}{n^2}\right) \tag{41}$$
$$= \hat{c}_{hy} h_k(\boldsymbol{x}_i, \boldsymbol{\theta}) + \hat{c}_y + \hat{c}_{yy}$$

Finally, factor the left-hand side of Equation 41 in terms of, $h_k(\boldsymbol{x}_i, \boldsymbol{\theta})$, 1, and $1^2$:

$$\underbrace{\left(c_{hy}\left(1 - \alpha \frac{n-1}{n}\right) - c_{hy}\frac{\alpha}{n}\right)}_{\hat{c}_{hy}} h_k(\boldsymbol{x}_i, \boldsymbol{\theta})$$
$$+ \underbrace{\left(c_y\left(1 - \alpha \frac{n-1}{n}\right) - c_y\frac{\alpha}{n}\right)}_{\hat{c}_y} + \underbrace{\left(c_{yy}\left(1 - \alpha \frac{n-1}{n}\right)^2 - c_{yy}\frac{\alpha^2}{n^2}\right)}_{\hat{c}_{yy}} \tag{42}$$

Thus, the in-parameterization constants with implicit label smoothing can be defined for any desired, label-smoothed constants as follows:

$$\hat{c}_1 = c_1 + c_y\frac{\alpha}{n} + c_{yy}\frac{\alpha^2}{n^2} \tag{43}$$

$$\hat{c}_h = c_h + c_{hy}\frac{\alpha}{n} \tag{44}$$

$$\hat{c}_{hh} = c_{hh} \tag{45}$$

$$\hat{c}_{hy} = c_{hy}\left(1 - \alpha \frac{n-1}{n}\right) - c_{hy}\frac{\alpha}{n} \tag{46}$$

$$\hat{c}_y = c_y\left(1 - \alpha \frac{n-1}{n}\right) - c_y\frac{\alpha}{n} \tag{47}$$

$$\hat{c}_{yy} = c_{yy}\left(1 - \alpha \frac{n-1}{n}\right)^2 - c_{yy}\frac{\alpha^2}{n^2} \tag{48}$$

So for any $\boldsymbol{\lambda}$ and any $\alpha \in (0, 1)$, there exists a $\hat{\boldsymbol{\lambda}}$ such that the behavior imposed by $\hat{\boldsymbol{\lambda}}$ without explicit label smoothing is identical to the behavior imposed by $\boldsymbol{\lambda}$ *with* explicit label smoothing. That is, any degree of label smoothing can be implicitly represented for any TaylorGLO loss function. Thus, TaylorGLO may discover and utilize label smoothing as part of discovering loss functions, increasing their ability to regularize further.

### D.3 SOFTMAX ENTROPY CRITICALITY

Let us analyze the case where all non-target logits have the same value, $\frac{\epsilon}{n-1}$, and the target logit has the value $1 - \epsilon$. That is, all non-target classes have equal probabilities.

A model's scaled logit for an input $\boldsymbol{x}_i$ can be represented as:

$$h_k(\boldsymbol{x}_i, \boldsymbol{\theta}) = \sigma_k(f(\boldsymbol{x}_i, \boldsymbol{\theta})) = \frac{e^{f_k(\boldsymbol{x}_i, \boldsymbol{\theta})}}{\sum_{j=1}^{n} e^{f_j(\boldsymbol{x}_i, \boldsymbol{\theta})}} \tag{49}$$

where $f_k(\boldsymbol{x}_i, \boldsymbol{\theta})$ is a raw output logit from the model.

The $(k, j)$th entry of the Jacobian matrix for $h(\boldsymbol{x}_i, \boldsymbol{\theta})$ can be easily derived through application of the chain rule:

$$\boldsymbol{J}_{kj} h(\boldsymbol{x}_i, \boldsymbol{\theta}) = \frac{\partial h_k(\boldsymbol{x}_i, \boldsymbol{\theta})}{\partial f_j(\boldsymbol{x}_i, \boldsymbol{\theta})} = \begin{cases} h_j(\boldsymbol{x}_i, \boldsymbol{\theta})\left(1 - h_k(\boldsymbol{x}_i, \boldsymbol{\theta})\right) f_k(\boldsymbol{x}_i, \boldsymbol{\theta}) & k = j \\ -h_j(\boldsymbol{x}_i, \boldsymbol{\theta}) h_k(\boldsymbol{x}_i, \boldsymbol{\theta}) f_k(\boldsymbol{x}_i, \boldsymbol{\theta}) & k \neq j \end{cases} \tag{50}$$

Consider an SGD learning rule of the form:

$$\theta_j \leftarrow \theta_j + \eta \frac{1}{n} \sum_{k=1}^{n} \left[\gamma_k(\boldsymbol{x}_i, \boldsymbol{y}_i, \boldsymbol{\theta}) D_{\boldsymbol{j}}\left(h_k(\boldsymbol{x}_i, \boldsymbol{\theta})\right)\right] \tag{51}$$

Let us freeze a network at any specific point during the training process for any specific sample. Now, treating all $f_j(\boldsymbol{x}_i, \boldsymbol{\theta}), j \in [1, n]$ as free parameters with unit derivatives, rather than as functions. That is, $\theta_j = f_j(\boldsymbol{x}_i, \boldsymbol{\theta})$. We observe that updates are as follows:

$$\Delta f_j \propto \sum_{k=1}^{n} \gamma_j \begin{cases} h_j(\boldsymbol{x}_i, \boldsymbol{\theta}) \left(1 - h_k(\boldsymbol{x}_i, \boldsymbol{\theta})\right) & k = j \\ -h_j(\boldsymbol{x}_i, \boldsymbol{\theta}) \, h_k(\boldsymbol{x}_i, \boldsymbol{\theta}) & k \neq j \end{cases} \tag{52}$$

For downstream analysis, we can consider, as substitutions for $\gamma_j$ above, $\gamma_{\neg T}$ to be the value for non-target logits, and $\gamma_T$ for the target logit.

This sum can be expanded and conceptually simplified by considering $j$ indices and $\neg j$ indices. $\neg j$ indices, of which there are $n - 1$, are either all non-target logits, or one is the target logit in the case where $j$ is not the target logit. Let us consider both cases, while substituting the scaled logit values defined above:

$$\Delta f_j \propto \begin{cases} \gamma_{\neg T} \, \boldsymbol{J}_{k=j} h(\boldsymbol{x}_i, \boldsymbol{\theta}) + (n-2)\gamma_{\neg T} \, \boldsymbol{J}_{k \neq j} h(\boldsymbol{x}_i, \boldsymbol{\theta}) + \gamma_T \, \boldsymbol{J}_{k \neq j} h(\boldsymbol{x}_i, \boldsymbol{\theta}) & \text{non-target } j \\ \gamma_T \, \boldsymbol{J}_{k=j} h(\boldsymbol{x}_i, \boldsymbol{\theta}) + (n-1)\gamma_{\neg T} \, \boldsymbol{J}_{k \neq j} h(\boldsymbol{x}_i, \boldsymbol{\theta}) & \text{target } j \end{cases} \tag{53}$$

$$\Delta f_j \propto \begin{cases} \begin{aligned} &\gamma_{\neg T} h_{\neg T}(\boldsymbol{x}_i, \boldsymbol{\theta}) \left(1 - h_{\neg T}(\boldsymbol{x}_i, \boldsymbol{\theta})\right) \\ &+ (n-2)\gamma_{\neg T} \left(-h_{\neg T}(\boldsymbol{x}_i, \boldsymbol{\theta}) \, h_{\neg T}(\boldsymbol{x}_i, \boldsymbol{\theta})\right) \\ &+ \gamma_T \left(-h_{\neg T}(\boldsymbol{x}_i, \boldsymbol{\theta}) \, h_T(\boldsymbol{x}_i, \boldsymbol{\theta})\right) \end{aligned} & \text{non-target } j \\ \\ \begin{aligned} &\gamma_T h_T(\boldsymbol{x}_i, \boldsymbol{\theta}) \left(1 - h_T(\boldsymbol{x}_i, \boldsymbol{\theta})\right) \\ &+ (n-1)\gamma_{\neg T} \left(-h_{\neg T}(\boldsymbol{x}_i, \boldsymbol{\theta}) \, h_T(\boldsymbol{x}_i, \boldsymbol{\theta})\right) \end{aligned} & \text{target } j \end{cases} \tag{54}$$

$$\text{where} \quad h_T(\boldsymbol{x}_i, \boldsymbol{\theta}) = 1 - \epsilon, \quad h_{\neg T}(\boldsymbol{x}_i, \boldsymbol{\theta}) = \frac{\epsilon}{n-1} \tag{55}$$

$$\Delta f_j \propto \begin{cases} \gamma_{\neg T} \dfrac{\epsilon}{n-1}\left(1 - \dfrac{\epsilon}{n-1}\right) + \gamma_{\neg T}(n-2)\dfrac{\epsilon^2}{n^2 - 2n + 1} + \gamma_T(\epsilon - 1)\dfrac{\epsilon}{n-1} & \text{non-target } j \\ \gamma_T \epsilon - \gamma_T \epsilon^2 + \gamma_{\neg T}(n-1)(\epsilon - 1)\dfrac{\epsilon}{n-1} & \text{target } j \end{cases} \tag{56}$$

At this point, we have closed-form solutions for the changes to softmax inputs. To characterize entropy, we must now derive solutions for the changes to softmax outputs given such changes to the inputs. That is:

$$\Delta \sigma_j(f(\boldsymbol{x}_i, \boldsymbol{\theta})) = \frac{\mathrm{e}^{f_j(\boldsymbol{x}_i, \boldsymbol{\theta}) + \Delta f_j}}{\sum_{k=1}^{n} \mathrm{e}^{f_k(\boldsymbol{x}_i, \boldsymbol{\theta}) + \Delta f_k}} \tag{57}$$

Due to the two cases in $\Delta f_j$, $\Delta \sigma_j(f(\boldsymbol{x}_i, \boldsymbol{\theta}))$ is thus also split into two cases for target and non-target logits:

$$\Delta \sigma_j(f(\boldsymbol{x}_i, \boldsymbol{\theta})) = \begin{cases} \dfrac{\mathrm{e}^{f_{\neg T}(\boldsymbol{x}_i, \boldsymbol{\theta}) + \Delta f_{\neg T}}}{(n-1)\mathrm{e}^{f_{\neg T}(\boldsymbol{x}_i, \boldsymbol{\theta}) + \Delta f_{\neg T}} + \mathrm{e}^{f_T(\boldsymbol{x}_i, \boldsymbol{\theta}) + \Delta f_T}} & \text{non-target } j \\ \dfrac{\mathrm{e}^{f_T(\boldsymbol{x}_i, \boldsymbol{\theta}) + \Delta f_T}}{(n-1)\mathrm{e}^{f_{\neg T}(\boldsymbol{x}_i, \boldsymbol{\theta}) + \Delta f_{\neg T}} + \mathrm{e}^{f_T(\boldsymbol{x}_i, \boldsymbol{\theta}) + \Delta f_T}} & \text{target } j \end{cases} \tag{58}$$

Now, we can see that scaled logits have a lower entropy distribution when $\Delta \sigma_T(f(\boldsymbol{x}_i, \boldsymbol{\theta})) > 0$ and $\Delta \sigma_{\neg T}(f(\boldsymbol{x}_i, \boldsymbol{\theta})) < 0$. Essentially, the target and non-target scaled logits are being repelled from each other. We can ignore either of these inequalities, if one is satisfied then both are satisfied, in part because $|\boldsymbol{\sigma}(f(\boldsymbol{x}_i, \boldsymbol{\theta}))|_1 = 1$. The target-case constraint (i.e., the target scaled logit must grow) can be represented as:

$$\frac{\mathrm{e}^{f_T(\boldsymbol{x}_i, \boldsymbol{\theta}) + \Delta f_T}}{(n-1)\mathrm{e}^{f_{\neg T}(\boldsymbol{x}_i, \boldsymbol{\theta}) + \Delta f_{\neg T}} + \mathrm{e}^{f_T(\boldsymbol{x}_i, \boldsymbol{\theta}) + \Delta f_T}} > 1 - \epsilon \tag{59}$$

Consider the target logit case prior to changes:

$$\frac{\mathrm{e}^{f_T(\boldsymbol{x}_i, \boldsymbol{\theta})}}{(n-1)\mathrm{e}^{f_{\neg T}(\boldsymbol{x}_i, \boldsymbol{\theta})} + \mathrm{e}^{f_T(\boldsymbol{x}_i, \boldsymbol{\theta})}} = 1 - \epsilon \tag{60}$$

Let us solve for $\mathrm{e}^{f_T(\boldsymbol{x}_i, \boldsymbol{\theta})}$:

$$\begin{aligned} \mathrm{e}^{f_T(\boldsymbol{x}_i, \boldsymbol{\theta})} &= (n-1)\mathrm{e}^{f_{\neg T}(\boldsymbol{x}_i, \boldsymbol{\theta})} + \mathrm{e}^{f_T(\boldsymbol{x}_i, \boldsymbol{\theta})} - \epsilon(n-1)\mathrm{e}^{f_{\neg T}(\boldsymbol{x}_i, \boldsymbol{\theta})} - \epsilon \mathrm{e}^{f_T(\boldsymbol{x}_i, \boldsymbol{\theta})} \tag{61} \\ &= \left(\frac{n-1}{\epsilon} - n + 1\right) \mathrm{e}^{f_{\neg T}(\boldsymbol{x}_i, \boldsymbol{\theta})} \tag{62} \end{aligned}$$

Substituting this definition into Equation 59:

$$\frac{e^{\Delta f_T}\left(\frac{n-1}{\epsilon} - n + 1\right)e^{f_{\neg T}(\boldsymbol{x}_i, \boldsymbol{\theta})}}{(n-1)e^{f_{\neg T}(\boldsymbol{x}_i, \boldsymbol{\theta}) + \Delta f_{\neg T}} + e^{\Delta f_T}\left(\frac{n-1}{\epsilon} - n + 1\right)e^{f_{\neg T}(\boldsymbol{x}_i, \boldsymbol{\theta})}} > 1 - \epsilon \quad (63)$$

Coalescing exponents:

$$\frac{e^{\Delta f_T + f_{\neg T}(\boldsymbol{x}_i, \boldsymbol{\theta})}\left(\frac{n-1}{\epsilon} - n + 1\right)}{(n-1)e^{f_{\neg T}(\boldsymbol{x}_i, \boldsymbol{\theta}) + \Delta f_{\neg T}} + e^{\Delta f_T + f_{\neg T}(\boldsymbol{x}_i, \boldsymbol{\theta})}\left(\frac{n-1}{\epsilon} - n + 1\right)} + \epsilon - 1 > 0 \quad (64)$$

Substituting in definitions for $\Delta f_T$ and $\Delta f_{\neg T}$ and greatly simplifying in a CAS is able to remove instances of $f_{\neg T}$:

$$\frac{\epsilon(\epsilon-1)\left(e^{\epsilon(\epsilon-1)(\gamma_{\neg T} - \gamma_T)} - e^{\frac{\epsilon(\epsilon-1)\gamma_T(n-1) + \epsilon\gamma_{\neg T}(\epsilon(n-3) + n - 1)}{(n-1)^2}}\right)}{(\epsilon-1)e^{\epsilon(\epsilon-1)(\gamma_{\neg T} - \gamma_T)} - \epsilon e^{\frac{\epsilon(\epsilon-1)\gamma_T(n-1) + \epsilon\gamma_{\neg T}(\epsilon(n-3) + n - 1)}{(n-1)^2}}} > 0 \quad (65)$$

## D.4 TaylorGLO parameter invariant at the null epoch

At the null epoch, a valid loss function aims to, in expectation, minimize non-target scaled logits while maximizing target scaled logits. Thus, we attempt to find cases of $\boldsymbol{\lambda}$ for which these behaviors occur. Considering the representation for $\gamma_k(\boldsymbol{x}_i, \boldsymbol{y}_i, \boldsymbol{\theta})$ in Equation 12:

$$\theta_j \leftarrow \theta_j + \eta\frac{1}{n}\sum_{k=1}^{n}\begin{cases}\left(c_1 + c_h h_k(\boldsymbol{x}_i, \boldsymbol{\theta}) + c_{hh}h_k(\boldsymbol{x}_i, \boldsymbol{\theta})^2\right)D_{\boldsymbol{j}}\left(h_k(\boldsymbol{x}_i, \boldsymbol{\theta})\right) & y_{ik} = 0 \\ \left(c_1 + c_h h_k(\boldsymbol{x}_i, \boldsymbol{\theta}) + c_{hh}h_k(\boldsymbol{x}_i, \boldsymbol{\theta})^2 \\ + c_{hy}h_k(\boldsymbol{x}_i, \boldsymbol{\theta}) + c_y + c_{yy}\right)D_{\boldsymbol{j}}\left(h_k(\boldsymbol{x}_i, \boldsymbol{\theta})\right) & y_{ik} = 1\end{cases} \quad (66)$$

Let us substitute $h_k(\boldsymbol{x}_i, \boldsymbol{\theta}) = \frac{1}{n}$ (i.e., the expected value of a logit at the null epoch):

$$\theta_j \leftarrow \theta_j + \eta\frac{1}{n}\sum_{k=1}^{n}\begin{cases}\left(c_1 + \frac{c_h}{n} + \frac{c_{hh}}{n^2}\right)D_{\boldsymbol{j}}\left(h_k(\boldsymbol{x}_i, \boldsymbol{\theta})\right) & y_{ik} = 0 \\ \left(c_1 + c_y + c_{yy} + \frac{c_h + c_{hy}}{n} + \frac{c_{hh}}{n^2}\right)D_{\boldsymbol{j}}\left(h_k(\boldsymbol{x}_i, \boldsymbol{\theta})\right) & y_{ik} = 1\end{cases} \quad (67)$$

For the desired degenerate behavior to appear, the directional derivative's coefficient in the $y_{ik} = 1$ case must be less than zero:

$$c_1 + c_y + c_{yy} + \frac{c_h + c_{hy}}{n} + \frac{c_{hh}}{n^2} \quad < \quad 0 \quad (68)$$

This finding can be made more general, by asserting that the directional derivative's coefficient in the $y_{ik} = 1$ case be less than $(n-1)$ times the coefficient in the $y_{ik} = 0$ case. Thus arriving at the following constraint on $\boldsymbol{\lambda}$:

$$c_1 + c_y + c_{yy} + \frac{c_h + c_{hy}}{n} + \frac{c_{hh}}{n^2} \quad < \quad (n-1)\left(c_1 + \frac{c_h}{n} + \frac{c_{hh}}{n^2}\right) \quad (69)$$

$$c_y + c_{yy} + \frac{c_{hy}}{n} \quad < \quad (n-2)\left(c_1 + \frac{c_h}{n} + \frac{c_{hh}}{n^2}\right) \quad (70)$$

The inverse of these constraints may be used as invariants during loss function evolution.

## E Examples of zero training error attractor dynamics

The strength of the attraction towards zero training error regions of the parameter space (described in Theorem 2) can be plotted—for any given number of classes $n$—at different $\epsilon$ values using the

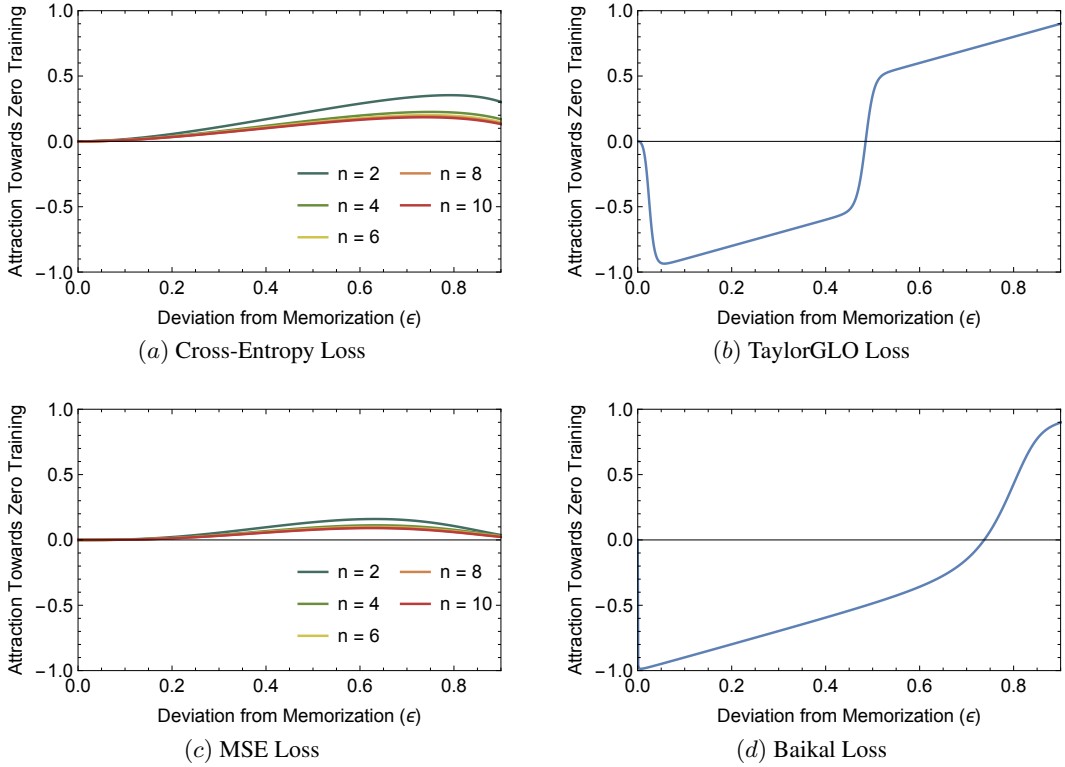

Figure 3: Attraction towards zero training error with different loss functions. Each loss function has a characteristic curve—plotted using Equation 17—that describes zero training error attraction dynamics for individual samples given their current deviation from perfect memorization, $\epsilon$. Plots (a) and (b) only have the $n = 10$ case plotted, i.e. the 10-class classification case for which they were evolved. Cross-entropy (a) and MSE (c) loss functions have positive attraction for all values of $\epsilon$. In contrast, the TaylorGLO loss function for CIFAR-10 on AllCNN-C (b) and the Baikal loss function (d) both have very strong attraction for weakly learned samples (on the right side), and repulsion for highly confidently learned samples (on the left side). Thus, this illustration provides a graphical intuition for the regularization that TaylorGLO and Baikal loss functions establish.

corresponding $\gamma_T$ and $\gamma_{\neg T}$ values from a particular loss function. These characteristic curves for four specific loss functions are plotted in Figure 3.

Both Baikal and TaylorGLO loss functions have significantly different attraction curves than the cross-entropy and mean squared error loss functions. Cross-entropy and mean squared error always exhibit positive attraction to zero training error. Conversely, TaylorGLO and Baikal only exhibit this positive attraction behavior for samples that are weakly memorized; well memorized samples produce a repulsive effect instead. This difference is what contributes to both metalearned loss functions' regularizing effects, where overconfidence is avoided.

## F   COMPARING MINIMA OF TRAINED NETWORKS

TaylorGLO loss functions are known to result in trained networks with flatter, lower minima (Gonzalez & Miikkulainen, 2020b). This result suggests that models trained with TaylorGLO loss function are more robust, i.e. their performance is less sensitive to small perturbations in the weight space, and that they also generalize better (Keskar et al., 2017). Since TaylorGLO loss functions that were discovered against an adversarial performance objective were even more robust (Section 6), what do their minima look like?

Model performance can be plotted along a random slice $[-1, 1]$ of the weight space using a prior loss surface visualization technique (Li et al., 2018). The random slice vector is normalized in a filter-wise

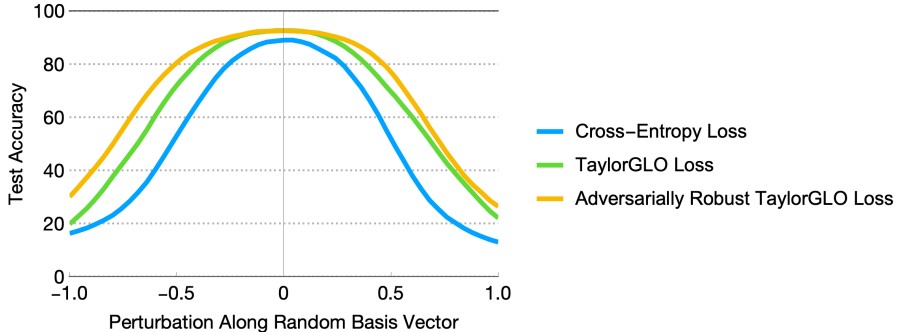

Figure 4: Comparing accuracy basins of AllCNN-C with cross-entropy, TaylorGLO, and adversarially robust TaylorGLO loss functions on CIFAR-10. Basins are plotted along only one perturbation direction for clarity, using a prior loss surface visualization technique (Li et al., 2018). While the adversarially robust TaylorGLO loss function leads to the same accuracy as the standard one, it has a wider, flatter minima. This result suggests that the TaylorGLO loss function that has been evolved to be robust against adversarial attacks is more robust in general, even when adversarial attacks are of no concern.

manner to accommodate network weights' scale invariance, thus ensuring that visualizations for two separate models can be compared. As a result of the randomness, this parameter space slice is unbiased and should take all parameters into account, to a degree. It can therefore be used to perturb trainable parameters systematically.

When AllCNN-C is trained with an adversarially robust versus a standard TaylorGLO loss function, its absolute accuracy is the same. However, the minimum is wider and flatter (Figure 4). This result suggests that it may be advantageous to evaluate TaylorGLO against an adversarial performance metric, even when the target application does not include adversarial attacks.

