# OpenReview forum: "Effective Regularization Through Loss-Function Metalearning"
_ICLR.cc/2021/Conference — Reject_

### Official Review · AnonReviewer1 · 2020-10-22
**With a few small adjustment, I think it is acceptable, with a few bigger adjustments I would favour it more.**

**Rating:** 7
**Confidence:** 3

**Review:**

UPDATE:

After the reviewers clarifications and some further explanations of the implications of Theorem 2 (in Appendix E) I think now that the paper tells an interesting story and thus I will vote to accept.



========================

Summary:

The paper addresses the setting of meta-learning loss functions and in particular analyses the effect of the loss function on the entropy of the resulting learned function. In particular it shows that TaylorGLO learned functions tend to lead to higher entropic, and thus more regularized, neural networks, than when they are trained with the cross entropy loss.  The paper also discusses that the property of high entropy predictions can lead to better robustness against adversarial attacks.

========================

Pros:

- Well written and structured, and thus easy to follow. (With a few exceptions, but I think with a bit effort that can be fixed. See additional feedback)

- Fairly unexplored but interesting setting.

========================

Cons:

- Some things are bit too informal, or not defined, see additional feedback.

- In my opinion the results are not very strong. In particular the one shown in Table 1. The result from Theorem 2 is to me weak in the sense that by itself it does not give any intuition in what is important for a loss function to reduce entropy, and what is important for the magnitude of it.  (See also additional feedback).

========================

Scoring:

For now I will vote for a weak accept, under the assumption that some of the smaller problems would be fixed for a final submission, see additional feedback. There you can also find what is missing for me for a stronger accept. I think the paper addresses an interesting and not much explored topic, and adds sufficient new insights to warrant a publication.

========================

========================

Additional feedback (along some questions.):

Recommendation for smaller adjustments:

- Third page first paragraph: ..'is important [for] the network's...'

- Introduce somewhere the Baikal and TaylorGLO loss, those are not that known.

- After Cross-Entropy analysis you refer to TaylorGLO's parameters a,b,c, which is at that point not introduced yet. Should somehow change the order.

- Below Theorem 1, there is a bracket that never opens.

- Theorem 2 is too informal. But in Theorem 2 you miss to introduce the gamma_T notation. For me both Theorem 1 and 2 rely too much on intuition, or are 'not attractors' and 'strength of entropy reduction' well-defined terms? (If so, then it should go to the appendix. As the intuition was at least clear to me, there is for me no strict need to change that though.)

- Under Table 1, you say that you use Theorem 2 to calculate the strength of the bias, but Theorem 2 only holds for the case in where all non-targets receive the same probability or not?

- Under Theorem 3, I don't understand why you need the inverse of the contraints to avoid non-usable loss functions.


Adjustments that would increase my score:

- Theorem 2 by itself is pretty void for me. I am missing that you draw some conclusions from it, at least for the loss functions you analyzed, in particular to analyze the magnitude of the bias.

- The results in Table 1 are essentially the same after adding the invariant, the experiment is not convincing to me. I think you should create maybe even a toy example where you really can highlight the potential benefit of the invariant.

- I do like the adversarial part. I would have found it very interesting to see how it compares to actually adversarially trained models. (I understand that this is not the point of the paper, but to me that would be still an interesting comparison)

=====================

---

> ### Author Response · Authors · 2020-11-23
> **Response**
>
> Dear Reviewer 1,
>
> Thank you for taking the time to review our paper. Below we address each of the concerns you raised.
>
> **RE: Small adjustments:**
> 	The paper is revised as suggested to resolve many of these great points that have been brought up.
>
> **RE: Theorem 2:**
> 	The expression defined in the theorem constitutes the basis for the attractor analyses in the paper. This connection is now made clear in the revision in Section 4.2, paragraphs 1 and 3.  Also, to make the conclusions concrete, specific instantiations of it for cross-entropy, TaylorGLO, Baikal, and MSE loss functions are now presented in Appendix E.
>
> **RE: Adversarial examples:**
> 	There is indeed interesting future work to be done in comparing to adversarially trained models, as well as finding specialized loss functions that improve their training. These directions are now mentioned in the last paragraph of Section 6.  Taking advantage of the additional page in the revision, we included new results for FGSM adversarial attacks to increases the impact of this section. Two new training settings are included in the main body, and new trained minima visualizations are shown in the appendix.
>
> We hope these clarifications and revisions to the paper have addressed your concerns.
>
> Best regards,
> – The Authors

---

> > ### Comment · AnonReviewer1 · 2020-11-24
> > **Response**
> >
> > Dear authors,
> >
> > thank you for the clarifications and additions. In particular the addition of Appendix E helps for me to get a better understanding what the implications of Theorem 2 are, at least for some well known loss functions.
> >
> > I am still wondering the following: Theorem 2 has the assumption that all non-target logits have the same probability. When you calculate the values for Figure 1 you use Theorem 2, but you can't verify that assumption? Or do I get something wrong here?
> >
> > Best Regards

---

> > > ### Author Response · Authors · 2020-11-25
> > > **Response**
> > >
> > > Dear Reviewer 1,
> > >
> > > Indeed, that assumption is applied to Figure 1. In previous studies, we found that the values for non-target logits are fairly similar in practice. The similarity increases as training progresses. Thus, we believe this seemingly imperfect application of Theorem 2 is valid, i.e. it is a good approximation from which conclusions may be drawn.
> > >
> > > Best regards,
> > > – The Authors

---

> > > > ### Comment · AnonReviewer1 · 2020-11-25
> > > > **Response**
> > > >
> > > > Thanks for the clarification! If you have space left that may be a worthwhile note for the reader.
> > > >
> > > > I still like the contribution of your paper and with the additions it conveys in my opinion an interesting story. I will thus up my vote for acceptance.

---

### Official Review · AnonReviewer3 · 2020-10-28
**Interesting approach but need more motivation for loss function learning**

**Rating:** 5
**Confidence:** 4

**Review:**

This paper analyzes a learned loss function called TaylorGLO based on third-order Taylor expansion and its regularization properties. This approach is novel and interesting in that the loss function is also learned on data. The analysis of the TaylorGLO loss and another learned loss function Baikal loss near zero error reveals interesting properties of preventing overconfident predictions.

However, I have some reservations on the idea of learning loss functions in general.
1. The improvements in classification accuracy using a learned loss function is relatively small, from previous works such as Gonzalez & Miikkulainen 2020b.
2. It is more difficult to interpret a learned loss function. What does a=-373.917, b=-129.928, c=-11.3145 mean in the Taylor expansion of the loss function?
3. Since the loss function is learned from data, would it become degenerate in the small sample regime?
4. Traditional loss functions like cross-entropy and hinge loss have classification rules that are Bayes consistent. It is not clear whether this is the case for learned loss functions.

Apart from these questions, there is also the issue of clarity in presentation. The authors should include some discussions or illustration of the Baikal or TaylorGLO loss in this paper to make it more self-contained.

As a question for Section 5, does the condition in Theorem 3 guarantee the trainability of the TaylorGLO loss (sufficient condition)? Or there are potentially other constraints needed?

Overall I believe this approach has merits in discovering new interesting functions for learning, followed by study of the loss function's properties by humans. But I am not convinced if we should directly use a learned loss function in training. I think there are more work needed for this paper before it could be accepted for ICLR.

---

> ### Author Response · Authors · 2020-11-23
> **Response**
>
> Dear Reviewer 3,
>
> Thank you for taking the time to review our paper. Below we address each of the main concerns you raised.
>
> **RE: Motivating loss-function metalearning and point #1:**
> 	With the additional page in the revised paper, we have now included a more thorough review of loss-function metalearning in Section 2.3. This prior work indeed already shows that loss-function metalearning improves performance. However, this paper focuses on the next goal: understanding why this is the case. It demonstrates theoretically that such metalearning establishes a process that reduces overfitting. This framing is now clearly expressed in Section 2.3 of the revision, in relation with prior work.
>
> **RE: 2. Definitions for a, b, and c:**
> 	The variables are defined in Appendix B and are now referenced in the body of the revised paper.
>
> **RE: 3. Metalearned loss functions in small sample regimes:**
> 	The prior GLO technique empirically showed that the Baikal loss function (which was metalearned) outperformed the cross-entropy loss on small training sets with as few as 250 samples.
>
> **RE: 4. Theoretical guarantees of metalearned loss functions:**
> 	Metalearned loss functions can be quite creative, and therefore do not necessarily have the same guarantees as e.g. cross-entropy. This work aims to provide a first step in this direction. Conceptual understanding is developed *a posteriori* with metalearned loss functions, rather than *a priori* as with the cross-entropy loss. This distinction is made clear in Section 2.3 in the revised paper.
>
> We hope these clarifications and revisions to the paper have addressed your concerns.
>
> Best regards,
> – The Authors

---

### Official Review · AnonReviewer4 · 2020-11-04
**Valuable & Novel Attractor Dynamics Evaluation**

**Rating:** 8
**Confidence:** 4

**Review:**

Summary:

Taylor polynomial based loss function metalearning acts as a regularizer that improves the networks adversarial attack robustness, performance, training time, and data utilization. The authors evolve weights, and so add arbitrary other factors to the loss, including adversarial robustness to learn a loss function parameterization which is more robust. They provide analysis of the attractor states under the optimization of a suite of loss functions.

Quality:

Writing Quality:
The authors have paid attention to detail. The writing is succinct and precise throughout, and I was only able to find one typo though the first 8 pages of the manuscript. The progression between concepts is well motivated.

Evaluation Quality:
The evaluation methodology is novel and its results address the dynamics of training rather than static outcomes. The choice of adversarial attack robustness to demonstrate the value of optimizing the loss function for an alternate metric is a sound one. Validation acuuracy is used elsewhere to evaluate the generalization ability of the loss. Transfer of the learned loss across datsets and models is not evaluated.

Result Quality:
The improvements in adversarial attack robustness are large when optimizing TaylorGLO directly for that objective. The differences in attractor dynamics are dramatic (but also are less surprising).

Clarity:

Graphs & Tables:
Figure 1, the attraction dynamics graph, is readable and quite clear. Unfortunately the entropy reduction definition of attraction in equation 17 can’t be efficiently described in the caption in the same way that it is described in the text.

Table 1’s invariance results are clearly presented. The Welch’s t-Test results generating P-value scores and the corresponding bolding is good. I would like to have seen a measure of meta-training stability or consistency in addition to or instead of accuracy, backing the claim about improved stability moving with the evolution population size.

Figure 2’s attack strength against testing accuracy plot is clean. Each axis’ meaning is clear, as and the interpretation of the result is natural.

The paper’s writing clarity is very good. The background is comprehensible. The decompositions in section 3 are well factored. The disagreement with Blanc et al. (2020) in section 4.1 can be fleshed out in more detail, but the writing in the rest of the section is precise and succinct.


Originality:

One challenge with addressing the originality in the paper is understanding what novelty should be attributed here and what should be attributed to the original TaylorGLO paper.

Novel evaluation methodology (attractor dynamics) are underemphasized relative to novel regularization results, and depends on the insight that the upside of the zero training error regime is that much of the continued update is all about the optimizer’s bias and not about the training data.


Significance:

One major question in this work concerns the generality of its findings. Are these attractor dynamics specific to TaylorGLO? What are ther implications for other regularizers?

The method’s added complexity makes the method unlikely to be used unless it can clearly differentiate itself from other regularizers. For example, label smoothing is likely to create very similar attractor dynamics to the dynamics seen in TaylorGLO. The regularization effect (output entropy penalty) is also very similar. One differentiating feature is the ability to add other objectives (like adversarial robustness) to the learned loss.


Pros:

The novel evaluation methodlogy which relies on the insight that all loss function changes will have a downstream impact on the gradients is a nice addition to the loss function metalearning suite. Attractor dynamics, optimizing for an alternate objective like adversarial robustness and the demonstrated flexibility of TaylorGLO to cover label smoothing, MSE, Cross-Entropy, and more are welcome contributions.

Cons:

The appreciation of the existing loss function metalearning literature is poor in this paper. Loss functions are commonly learned with Neural Networks! These losses are often easier to optimize and are more flexible than standard losses. They can also make non-differentiable feedback differentiable. See this metalearning survey for plenty of references. https://arxiv.org/pdf/2004.05439.pdf

Comparisons between taylor approximation paremeterized loss functions and neural network parameterized loss functions would have been an important comparison to see, but this side of loss function metaleanring isn’t referenced. What are the attractor dynamics for those neural network learned losses? Is this different / improved? While many of these papers focus on reinforcement learning or unsupervised learning, they point to very similar improvements coming out of loss function metalearning. Ex, any of the following:

Evolved Policy Gradient
https://arxiv.org/abs/1802.04821
Learning to Learn: Meta-Critic Networks for Sample Efficient Learning
https://www.researchgate.net/publication/318029457_Learning_to_Learn_Meta-Critic_Networks_for_Sample_Efficient_Learning
Online Meta-Critic Learning for Off-Policy Actor-Critic Methods
https://arxiv.org/abs/2003.05334
Online-Within-Online Meta-Learning (learned regularization)
http://papers.nips.cc/paper/9468-online-within-online-meta-learning.pdf
Meta-Learning Update Rules for Unsupervised Representation Learning
https://arxiv.org/abs/1804.00222
Learning to Learn by Self-Critique
https://papers.nips.cc/paper/9185-learning-to-learn-by-self-critique


This paper doesn’t focus on task generality. Many of the other metalearning loss functions papers do. Why? How general are their learned losses? Can they generalize from task to task, or does it have to be retrained every time? Why don’t they discuss these issues?

They don’t release code for reproducibility.


Notes:

Ideally Figure 1’s information would be shown for Bikal, MSE, and Label Smoothing as well (perhaps in the appendix) to assess whether TaylorGLO’s training dynamics add anything on top of Label Smoothing (which one would expect to have the same transition to push away from the correct label in later epochs). But there the definition of zero training error itself is modified, and so their metric may not capture very similar optimization dynamics.

The claim that TaylorGLO lowers confidence could be evaluated on calibration, rather than or in addition to entropy.

TaylorGLO may be doing much more than regularization in practice, and the evaluiton criteria don’t seem sufficient to know that more isn’t happening to the model optimzed for this loss.

It would be good to see the attractor dynamic graphs for label smoothing (presumably it is very similar to TaylorGLO).

It appears that Theorem 4.2 basically describes label smoothing, though they don’t say this.

A typo on page 6! “Thus, values less than zero imply that entropy is increased, values greater than zero that it is decreased, and values equal to zero imply that there is no change.”

---

> ### Author Response · Authors · 2020-11-23
> **Response**
>
> Dear Reviewer 4,
>
> Thank you for taking the time to review our paper. Below we address each of the main concerns you raised.
>
> **RE: Transfer of the learned loss across datasets and models:**
> 	Transfer is indeed not a focus of this paper—it focuses on understanding the generalization properties of loss-function metalearning instead.  However, this question was addressed in prior work (Gonzalez and Miikkulainen 2020). They demonstrated that while learned loss-functions transfer across datasets and models to some extent, they are most powerful when they are customized to individual tasks and architectures, ostensibly by taking advantage of the different characteristics of each such setting.  This point is now made in the review of prior work in Section 2.3.
>
> **RE: Clarifications:**
> 	The paper is revised as suggested to resolve many of these great points that have been brought up.
>
> **RE: Generality of findings:**
> 	Many of the analyses focus on TaylorGLO because it is the most recent and scalable of the loss-function metalearning methods.  However, the conclusions about the nature of regularization and what role loss functions can play in it are general and apply to other methods, such as GLO. We have revised the paper to make this point clear. In addition, as a concrete example of this generality, an analysis of the attractor dynamics of Baikal, a loss function discovered by a loss-function metalearning technique different from TaylorGLO, is now included in Appendix E.
>
> **RE: Code for reproducibility:**
> 	If accepted, we will be publicly open-sourcing code for performing the calculations and analyses shown in the paper.
>
> **RE: A measure of meta-training stability or consistency for invariant experiments:**
> 	This is a great suggestion and we are working on running experiments for it now; while it wasn’t possible to finish by the end of the rebuttal period, we will include those results in the final version of the paper. We are confident that we will be able to provide stability metrics.
>
> **RE: Loss-function metalearning literature:**
> 	With the additional page in the revised paper, we have now included a thorough literature review in Section 2, including all the papers suggested by the Reviewer. This prior work indeed already shows that loss-function metalearning improves performance. However, this paper focuses on the next goal: understanding why this is the case. It demonstrates theoretically that such metalearning establishes a process that reduces overfitting. This framing is now clearly expressed in Section 2.3 of the revision, in relation with prior work.
>
> We hope these clarifications and revisions to the paper have addressed your concerns.
>
> Best regards,
> – The Authors

---

### Official Review · AnonReviewer5 · 2020-11-09
**More work needs to be done**

**Rating:** 3
**Confidence:** 3

**Review:**

This paper mainly deals with the theoretical support for the loss function meta-learning and focuses on illustrating the generalization superiority of the TaylorGLO method [1]. Although the generalization performance is the core aspect of machine learning, I have several concerns as follows.

1. Since the TaylorGLO method [1] is also under review in ICLR 2021, I can not judge the value of this paper.
2. This paper claims that it theoretically analyzes the generalization superiority of the TaylorGLO method. But I cannot get this point from this paper. In my opinion, when we consider the generalization performance, the generalization error bound is preferred to answer this. Here the authors try to analyze the training dynamics of SGD for deep models since previous work has shown the implicit regularization effect of the gradient-based optimization algorithms. But, what's the connection between the training dynamics and generalization, or what's the hypothesis between these two in this paper? Do I miss something?
3. what do the theorems provided in this paper want to tell? I don't get the points that the authors want to tell. More intuitive explanations should be given followed by the theorem.


[1] OPTIMIZING LOSS FUNCTIONS THROUGH MULTI-VARIATE TAYLOR POLYNOMIAL PARAMETERIZATION, https://openreview.net/pdf?id=bJLHjvYV1Cu

---

> ### Author Response · Authors · 2020-11-23
> **Response**
>
> Dear Reviewer 5,
>
> Thank you for taking the time to review our paper. We would like to clarify some of your concerns:
> 1. While both papers involve TaylorGLO, they each make an independent contribution and should be evaluated separately. Whereas reference [1] presents an experimental analysis of TaylorGLO, this paper focuses on understanding the regularization effect of loss-function metalearning in general. TaylorGLO is used as the basis for many of the analyses because it is the most recent and scalable of such methods, but it is still only a vehicle to obtain general insights into the nature of regularization and what role loss functions can play in it. The same conclusions apply to other methods, such as GLO. We have revised the paper to make this focus clear (Section 2.3). In addition, as a concrete example of the generality, we have included further analyses of Baikal, a loss function discovered by a loss-function metalearning technique different from TaylorGLO, in the paper (Appendix E).
> 2. The goal of the paper is actually slightly different. The point is not to demonstrate that TaylorGLO has superior generalization, but instead to understand where the generalization ability comes from. Prior work has already demonstrated that loss-function metalearning improves generalization; this paper shows theoretically how such learning in TaylorGLO and GLO results in dynamics that keeps the networks from overfitting. It therefore establishes an explanation for the observed effect—an insight which then leads to further improvements of these methods. The paper has been revised to make this focus and connection clear (Section 2.3, last paragraph).
> 3. The paper has been revised to provide clearer motivations and interpretations of the theorems. These revisions also respond to concern #2, i.e. to make it clear how the theorems support the goal of the paper.
>
> We hope these clarification and the updates to the paper address your concerns.
>
> Best regards,
> – The Authors

---

> ### Comment · AnonReviewer1 · 2020-11-25
> **Response**
>
> Dear Reviewer 5,
>
> after the clarifications and additions of the authors I think that the paper is well rounded and tells an interesting story. Did any of their clarifications change your mind?
>
> Regarding your concerns of the paper. I disagree with the statement that "This paper claims that it theoretically analyzes the generalization superiority of the TaylorGLO method.". This indeed cannot be followed from the paper, and reading it under that assumption I understand that the Theorems do not support this claim. The paper rather starts from the assumption the Baikal loss and TaylorGLO do improve generalization (at least in some instances.) That makes sense to me as those loss functions are found via meta-learning, so actually are informed choices, and the assumption is backed up (at least for the Baikal loss) with references.
>
> The question this paper asks instead: What are the inherent differences of those losses to the classical MLE and Cross entropy loss, and this question is answered from the perspective of the attractor dynamics.
>
> Also the outlook on adversarial robustness is a very interesting direction. The losses are found for optimal generalization properties, but seem to be more adversarial robust.
>
> Best Regards

---

### Author Response · Authors · 2020-11-23
**Revision Overview**

We would like to thank all the reviewers for their time and effort in reading the paper and for their constructive comments. We have addressed the main concerns and updated the paper. The major updates are:

- New Experiments and Analyses:
    - New adversarial attack robustness experiments on Wide ResNet 16-8 and AllCNN-C with Cutout
    - Minima visualizations for adversarial attack experiments
    - Attraction towards zero training error curves for the cross-entropy, TaylorGLO, Baikal, and MSE loss functions
- Clarifications:
    - More in-depth literature review that covers loss-function metalearning in greater detail and more clearly gives context to the contributions in this paper
    - Small clarifications and improvements throughout

We have also responded to each reviewer’s specific comments in the replies to their reviews.

---

### Decision · Program_Chairs · 2021-01-07
**Final Decision**

**Decision:**

Reject

**Comment:**

The paper presents a method for meta-learning the loss function. The analysis mainly concerns the recently proposed TaylorGLO method on the (slightly less recent) Baikal loss. There was no consensus on this paper, but no reviewer was willing to fight for acceptance either. I found the paper not self-contained, with important non-standard elements undefined, starting with the Baikal loss, notations that are not defined in the main text, and a nomenclature that is also unusual with important terms such as "attractor" or "invariant" used in meanings that are non-standard in optimization or machine learning.

Regarding content, most of the analyses refer to properties of the Baikal loss (not presented in the main text) that are deemed to be positive, without any theoretical support (Theorems 1 and 2). The inability to overfit is here posed as an obvious quality of a training loss. Then, a way to prevent the failure of the meta-training algorithm is presented in Theorem 3. Finally, an experiment is provided, showing that the proposed meta-training algorithm performs better than "vanilla" training with respect to adversarial attacks with FGSM. There is no comparison with other defense mechanisms and no analysis explains the results. Overall, although some interesting aspects may be developedin this paper, they are currently not well served by writing or the experimental evidences, so I recommend rejection.